# Divergent midbrain circuits orchestrate escape and freezing responses to looming stimuli in mice

Congping Shang[1,2,3,4], Zijun Chen[1,2], Aixue Liu[4,5], Yang Li[4], Jiajing Zhang[1], Baole Qu[1,2], Fei Yan[1,2], Yaning Zhang[3,4], Weixiu Liu[3,4], Zhihui Liu[1], Xiaofei Guo[1], Dapeng Li[4], Yi Wang[1] & Peng Cao [1,4]

Animals respond to environmental threats, e.g. looming visual stimuli, with innate defensive behaviors such as escape and freezing. The key neural circuits that participate in the generation of such dimorphic defensive behaviors remain unclear. Here we show that the dimorphic behavioral patterns triggered by looming visual stimuli are mediated by parvalbumin-positive (PV$^+$) projection neurons in mouse superior colliculus (SC). Two distinct groups of SC PV$^+$ neurons form divergent pathways to transmit threat-relevant visual signals to neurons in the parabigeminal nucleus (PBGN) and lateral posterior thalamic nucleus (LPTN). Activations of PV$^+$ SC-PBGN and SC-LPTN pathways mimic the dimorphic defensive behaviors. The PBGN and LPTN neurons are co-activated by looming visual stimuli. Bilateral inactivation of either nucleus results in the defensive behavior dominated by the other nucleus. Together, these data suggest that the SC orchestrates dimorphic defensive behaviors through two separate tectofugal pathways that may have interactions.

[1] State Key Laboratory of Brain and Cognitive Sciences, Institute of Biophysics, Chinese Academy of Sciences, 100101 Beijing, China. [2] University of Chinese Academy of Sciences, 100049 Beijing, China. [3] PTN Graduate Programs, School of Life Sciences, Tsinghua University, 100084 Beijing, China. [4] National Institute of Biological Sciences, 102206 Beijing, China. [5] Graduate School of Peking Union Medical College, Chinese Academy of Medical Sciences, 100730 Beijing, China. These authors contributed equally: Congping Shang, Zijun Chen. Correspondence and requests for materials should be addressed to P.C. (email: caopeng@nibs.ac.cn)

Innate defensive behaviors triggered by environmental threats play a critical role in animal survival[1,2]. Among these behaviors, fleeing and freezing are fundamental forms in natural and laboratory conditions[3,4]. For example, in response to looming visual stimuli mimicking a predator, rodents[5–7] and *Drosophila*[8,9] exhibit defensive behaviors with either escape or freezing patterns. The generation of appropriate behavioral pattern to defend against threats is a demanding task across species[7,10–13]. How the brain detects threats and coordinates the generation of either fleeing or freezing is an unresolved question[14–16].

In rodent brain, the superior colliculus (SC) is a retinal recipient structure[17–19] involved in visual information processing[20–23], sensorimotor transformation[24], and cognitive functions[25]. In addition to mediating orienting responses[26], the rodent SC strongly contributes to triggering defensive behaviors[27–31]. Notably, early seminal studies of the SC indicated that distinct defensive behavioral patterns can be triggered by stimulation at different sites in the SC[27]. These studies were followed by the identification of looming-sensitive cells in specific layers of the SC[32]. These behavioral and physiological data suggest that, among the intermingled diverse cell types in the SC[21,33], a subset of neurons may detect looming visual stimuli and participate in the generation of dimorphic defensive behaviors. However, the identity of these looming detectors and their downstream neural circuits to orchestrate innate defensive behaviors is poorly defined.

Recently, it has been shown that SC neurons expressing parvalbumin (PV$^+$) may be a key neuronal subtype to trigger stereotyped defensive behaviors[34]. However, it is unknown whether the SC PV$^+$ neurons participate in coordinating the action selection between fleeing and freezing. In the present study, we systematically examined the roles of these neurons and their divergent downstream pathways in visually triggered defensive behaviors. Our data indicate that the SC PV$^+$ neurons orchestrate these dimorphic defensive behaviors with two divergent tectofugal visual pathways.

## Results

### Quantification of visually triggered defensive behaviors.
We measured defensive behaviors triggered by overhead looming visual stimuli mimicking an approaching predator[5] in an open field using 58 wild-type (WT) adult male mice naïve to the looming visual stimuli (Fig. 1a). To avoid potential behavioral bias caused by animal gender[34], we used male mice in the present study. Mouse behavior was recorded by two orthogonally positioned cameras and analyzed off-line (Supplementary Fig. 1a, b). The looming visual stimuli (single trial), consisting of three cycles of an expanding dark disk that last 3 s (Fig. 1b), triggered either impulsive escape followed by freezing (Video 1) or immediate freezing (Video 2). To quantitatively describe these behavioral patterns, we measured the average locomotion speed before (3 s), peak speed during (3 s), and average speed after (15 s) looming visual stimuli (Fig. 1c, d). We then calculated the locomotion speed index of each mouse during stimuli (LSI$_{during\ stimuli}$) and after stimuli (LSI$_{after\ stimuli}$) (Fig. 1e). LSI$_{during\ stimuli}$ was calculated as the log (base of 10) of the ratio between peak speed during stimuli (3 s) and average speed before stimuli (3 s); LSI$_{after\ stimuli}$ was calculated as the log of the ratio between average speed after stimuli (15 s) and that before stimuli (3 s). Defensive behavior with a positive LSI$_{during\ stimuli}$ value and negative LSI$_{after\ stimuli}$ value was defined as an escape-freezing pattern (Type I), whereas that with negative values for both LSI$_{during\ stimuli}$ and LSI$_{after\ stimuli}$ was defined as a freezing-only pattern (Type II). Accordingly, most mice (44/58, ~76%) showed the Type I pattern (escape: latency = 154 ± 32 ms, duration = 0.52 ± 0.09 s; freezing:

duration = 28 ± 4.5 s; Fig. 1f, h). A small proportion of mice (14/58, ~24%) showed the Type II pattern (freezing: latency = 163 ± 29 ms; duration = 21 ± 5.3 s; Fig. 1g, i). These data indicate that looming visual stimuli trigger stereotyped dimorphic defensive behavioral patterns in mice.

In response to repeated looming visual stimuli (3 trials, 1 trial per minute), the same mice consistently exhibited the same behavioral pattern (Supplementary Fig. 1c, d). However, the LSI$_{during\ stimuli}$ and LSI$_{after\ stimuli}$ of mice with Type I or Type II behavioral pattern rapidly declined (Supplementary Fig. 1c, d), reflecting an adaptation to the repeated trials. Thus, to avoid behavioral adaptation caused by repeated trials, we applied only one trial for each mouse in the subsequent experiments.

### SC PV$^+$ neurons essential for dimorphic defensive behaviors.
The SC has been implicated in visually triggered defensive behaviors in rodents[26,30,32]. Among different subtypes of SC neurons, PV$^+$ neurons are sufficient to trigger defensive behaviors in mice[34]. These neurons are predominantly distributed in the retinal recipient layers of the SC, i.e. the superficial gray layer (SuG) and optic nerve layer (Op)[34]. Before exploring the role of SC PV$^+$ neurons in dimorphic defensive behaviors, we examined their neurotransmitter type by immunostaining of endogenous PV together with glutamate or GABA, the antibodies of which have been validated in this study (Supplementary Fig. 2). In the retinal recipient layers of the SC, most PV$^+$ neurons were immunohistochemically positive for glutamate (84% ± 8%, n = 5 mice; Fig. 2a) and negative for GABA (82% ± 9%, n = 5 mice; Fig. 2a). For cell counting strategy, see Methods and Supplementary Table 1. In other mammalian species, SC PV$^+$ neurons rarely colocalize with GABA[35,36], supporting our observation in mice. These morphological data, together with published physiological evidence[34], indicate that most SC PV$^+$ neurons in mice are glutamatergic.

Then we explored the role of SC PV$^+$ neurons in visually triggered Type I and Type II defensive behaviors. Tetanus neurotoxin (TeNT), a protease to block neurotransmitter release by cleaving synaptobrevin-2, has been widely used as a molecular tool for neuronal inactivation[37,38]. To inactivate SC PV$^+$ neurons, we injected adeno-associated virus (AAV) expressing double-floxed EGFP and TeNT (AAV-DIO-EGFP-2A−TeNT) into the bilateral SC of *PV-ires-Cre* mice (Supplementary Fig. 3a). This approach resulted in specific expression of EGFP in SC PV$^+$ neurons (Fig. 2b and Supplementary Fig. 3b), without retrogradely labeling cells in the substantia nigra pars reticulata (SNr) or retina (Supplementary Fig. 3c−f). To test the efficiency of TeNT-induced synaptic inactivation, we injected a mixture of AAV-DIO-EGFP-2A-TeNT and AAV-DIO-ChR2-mCherry unilaterally into the SC of *PV-ires-Cre* mice, resulting in the co-expression of EGFP and ChR2-mCherry in the same PV$^+$ neuron (Fig. 2c and Supplementary Fig. 3g). The amplitude of light-evoked post-synaptic currents (PSCs) from ChR2-mCherry-negative cells in SuG and Op layers of acute SC slices was strongly reduced by the expression of TeNT in SC PV$^+$ neurons (Fig. 2d, e), suggesting that neurotransmitter release from SC PV$^+$ neurons was effectively blocked. TeNT expression for 3 weeks did not alter the resting membrane potential, intrinsic neuronal excitability, suggesting the effect of TeNT is specific to neurotransmitter release (Supplementary Fig. 3h, i). Cleaved caspase 3, an apoptotic marker, was not observed in SC PV$^+$ neurons even 3 months after AAV injection, suggesting that TeNT expression has minimal effect on the health of SC PV$^+$ neurons (Supplementary Fig. 3j−l). Finally, in SC PV$^+$ neurons infected by mixed AAV-DIO-ChR2-mCherry and AAV-DIO-EGFP-2A-TeNT, TeNT expression did not impair light-evoked action potential firing (Supplementary Fig. 3, m-o), supporting

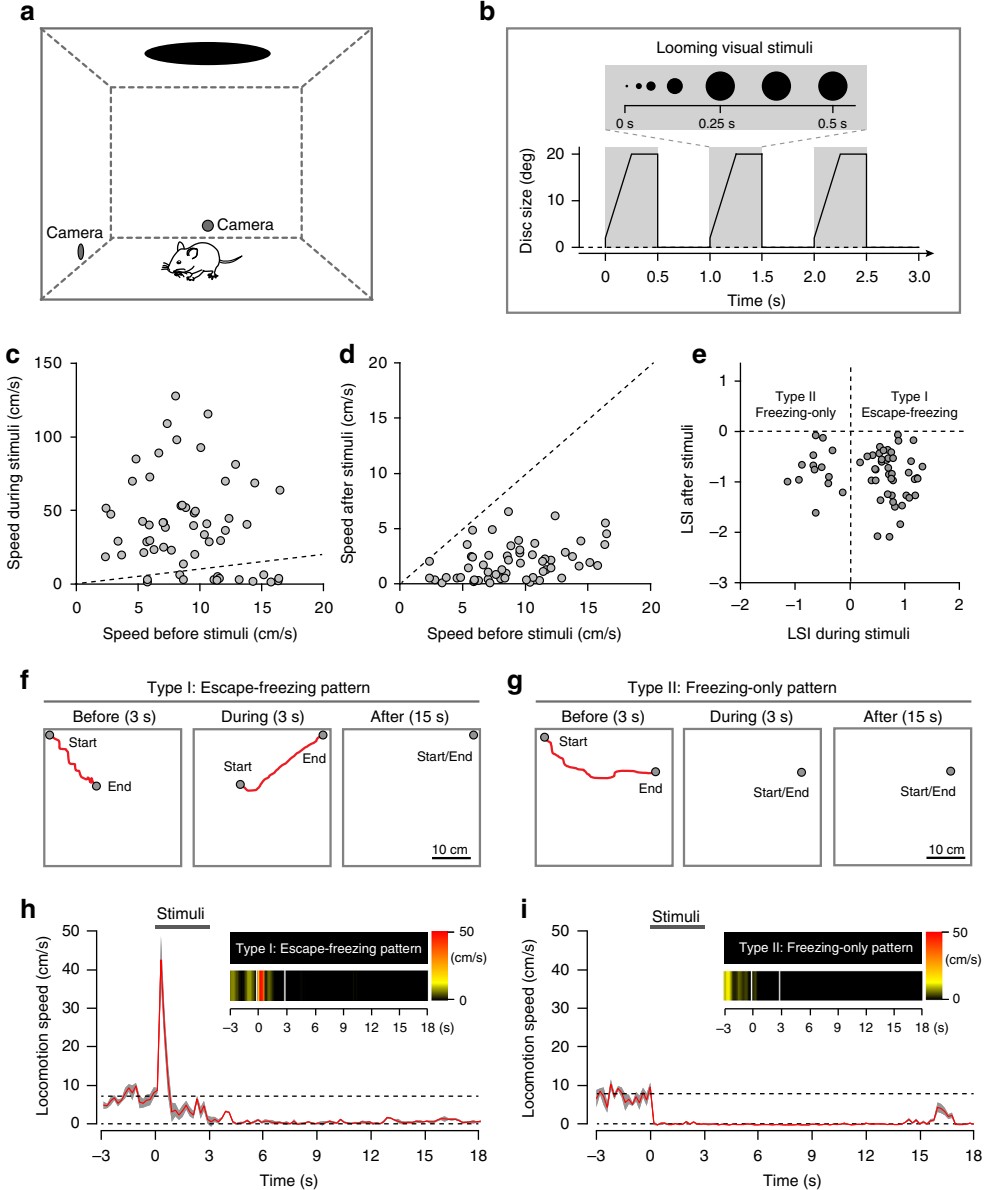

**Fig. 1** Quantitative analyses of mouse defensive behaviors triggered by looming visual stimuli. **a** Schematic diagram showing the arena where a mouse received overhead looming visual stimuli. Locomotion was recorded by two orthogonally positioned cameras. **b** A dark disk expanding from 2 to 20 degrees for three repeated cycles was used as the looming visual stimulus to trigger defensive behaviors. **c** Distribution of average locomotion speed 3 s before visual stimuli (for abbreviation, speed before stimuli) and peak speed during visual stimuli (speed during stimuli) of 58 male mice. **d** Distribution of average locomotion speed 3 s before visual stimuli (speed before stimuli) and average speed 15 s after stimuli (speed after stimuli) of 58 male mice. **e** Distribution of LSI$_{during\ stimuli}$ and LSI$_{after\ stimuli}$ of 58 male mice showing two defensive behavioral patterns. **f, g** Locomotion trails from example mice with Type I (**f**) and Type II (**g**) defensive behavioral patterns before (3 s), during (3 s), and after visual stimuli (15 s). **h, i** Summarized time course of locomotion speed of mice with Type I (h, $n = 44$ mice) and Type II (I, $n = 14$ mice) defensive behavioral patterns. Dashed lines, the averaged locomotion speed before looming visual stimuli. Gray cloudy area indicates SEM of the averaged data. Inset, locomotion speed heat-maps of example mice

that the reduction in light-evoked PSCs (Fig. 2d, e) was due to synaptic inactivation.

Then we evaluated the effects of SC PV$^+$ neuron inactivation on visually triggered defensive behaviors (Video 3), by measuring the locomotion speed before, during, and after visual stimuli of mice with active (Ctrl: AAV-DIO-EGFP) and inactive (TeNT: AAV-DIO-EGFP-2A-TeNT) SC PV$^+$ neurons (Fig. 2f, g). The distribution of LSI$_{during\ stimuli}$ and LSI$_{after\ stimuli}$ indicated that the proportions of the two types of behavioral patterns (Ctrl: Type I/Type II = 19/8; TeNT: Type I/Type II = 17/10) were barely altered by TeNT expression in SC PV$^+$ neurons (Fig. 2h). For mice with Type I behavioral pattern (Ctrl: $n = 19$ mice;

TeNT: $n = 17$ mice), TeNT mice exhibited reduced escape speed during stimuli and increased speed during freezing phase after stimuli, compared with Ctrl mice (Fig. 2i, j). For mice with Type II behavioral pattern (Ctrl: $n = 8$ mice; TeNT: $n = 10$ mice), TeNT mice showed increased locomotion speed both during and after visual stimuli (Fig. 2i, k). These data indicate that, as a key neuronal subtype in SC circuits, PV$^+$ neurons are essential for both Type I and Type II defensive behavioral patterns.

**SC PV$^+$ neurons project to multiple brain areas**. We next explored how SC PV$^+$ neurons mediate both Type I and Type II

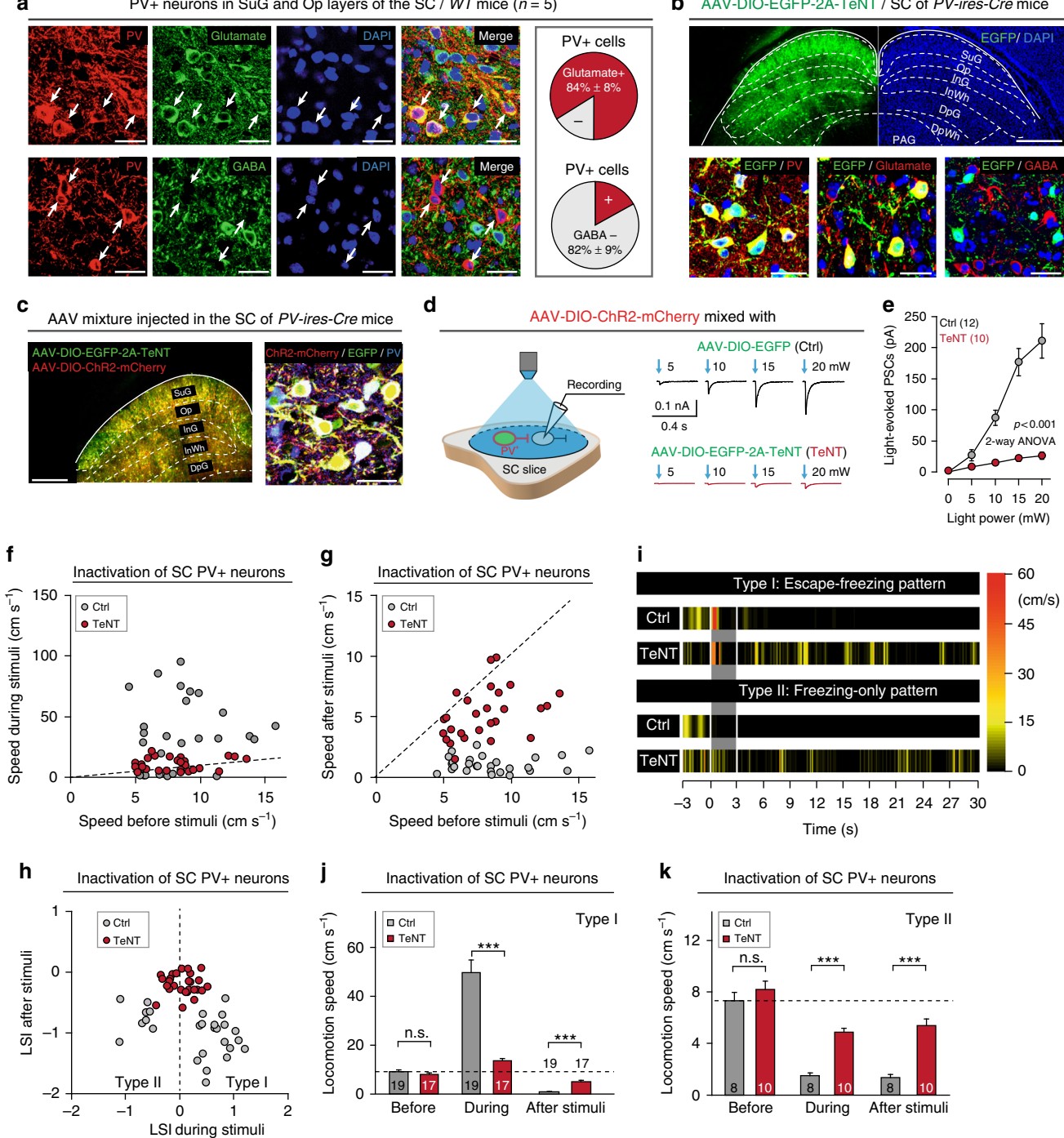

**Fig. 2** SC PV$^+$ neurons are essential for visually triggered dimorphic defensive behaviors. **a** Left, Micrographs showing immunostaining of PV vs. glutamate (top) or PV vs. GABA (bottom) in SuG layer of the SC. Arrows indicate PV$^+$ neurons. Right, quantitative analyses indicating most SC PV$^+$ neurons in SuG and Op layers are glutamate$^+$ and GABA$^-$. Scale bars, 30 μm (top) and 35 μm (bottom). **b** Coronal SC section (top) and micrographs (bottom) of *PV-ires-Cre* mice injected with AAV-DIO-EGFP-2A-TeNT, showing expression of EGFP in SC neurons positive for PV, glutamate but negative for GABA. Scale bars, 0.5 mm (top) and 30 μm (bottom). **c** Coronal SC section (left) and micrograph (right) showing SC PV$^+$ neurons co-labeled by EGFP and ChR2-mCherry after injection of AAV-DIO-EGFP-2A-TeNT and AAV-DIO-ChR2-mCherry mixture into the SC of *PV-ires-Cre* mice. Scale bars, 0.3 mm (left) and 20 μm (right). **d** Schematic diagram of slice physiology (left) and example traces (right) showing neurotransmitter release from SC PV$^+$ neurons was blocked by TeNT. **e** Input−output curves of light-evoked PSCs recorded from ChR2-negative neurons. **f, g** Distribution of speed during vs. before (**f**) and after vs. before (**g**) visual stimuli of mice with active (Ctrl) or inactive (TeNT) SC PV$^+$ neurons. **h** Distribution of LSI$_{during\ stimuli}$ and LSI$_{after\ stimuli}$ of mice with active (Ctrl) or inactive (TeNT) SC PV$^+$ neurons. **i** Heat-maps of speed time course of example mice with active (Ctrl) or inactive (TeNT) SC PV$^+$ neurons. **j, k** Effects of synaptic inactivation of SC PV$^+$ neurons on locomotion speed of mice with Type I (**j**) and Type II (**k**) defensive behavioral patterns. For single-channel micrographs (**b**, **c**), see Supplementary Fig. 3. Data in (**e**, **j**, **k**) are means ± SEM (error bars). Numbers of cells or mice are indicated in the graphs. Statistical analysis was Student's *t* test (***$P < 0.001$; n.s. $P > 0.1$) and one-way ANOVA

defensive behavioral patterns. Previous study[34] reported that activation of the pathway from SC PV$^+$ neurons to the parabigeminal nucleus (PBGN) triggered immediate escape followed by freezing, resembling Type I behavioral pattern in the present study. This observation, together with the above result that SC PV$^+$ neurons are essential for both Type I and Type II defensive behavioral patterns, raised a possibility that SC PV$^+$ neurons may form additional, yet unidentified, pathways associated with Type II defensive behavioral pattern. This prompted us to perform a whole-brain mapping to systematically search for the downstream targets of SC PV$^+$ neurons.

By injecting AAV-DIO-mGFP into the SC of *PV-ires-Cre* mice, we selectively labeled SC PV$^+$ neurons with a membrane-bound form of GFP (mGFP; Fig. 3a, b). In addition to confirming the projection areas (PBGN, Pn, and DLGN) reported previously[34], the whole-brain mapping identified six additional brain regions as downstream targets of SC PV$^+$ neurons (Supplementary Fig. 4a, b). Notably, the lateral posterior thalamic nucleus (LPTN), which has been shown to project to the lateral amygdala to trigger innate fear responses[39,40], was found to receive a projection from the SC PV$^+$ neurons (Fig. 3c, d). Thus, these anterograde tracing data suggest PV$^+$ SC-PBGN and PV$^+$ SC-LPTN as two strong candidate pathways to mediate the visually triggered dimorphic defensive behaviors.

To further characterize the PV$^+$ SC-PBGN and PV$^+$ SC-LPTN pathways, we performed retrograde tracing with cholera toxin B (CTB) tagged with Alexa Fluor-488 (CTB-488) and Alexa Fluor-594 (CTB-594). Both CTB-488 and CTB-594 were injected into the ipsilateral PBGN and LPTN of the same mice, respectively (Fig. 3e). The SC neurons retrogradely labeled with CTB-488 were broadly distributed in different layers, whereas those labeled with CTB-594 were clustered in the Op layer (Fig. 3f and Supplementary Fig. 4c). In the SuG and Op layers, a large proportion of PBGN-projecting (CTB-488$^+$, 56% ± 6.3%, $n = 3$ mice) and LPTN-projecting SC neurons (CTB-594$^+$, 66% ± 8.1%, $n = 3$ mice) were positive for PV (Fig. 3g). Among the SC PV$^+$ neurons labeled by CTB-488 or CTB 594, only a small proportion of them were dually labeled (Fig. 3h), indicating that very few SC PV$^+$ neurons have collaterals projecting to both the PBGN and LPTN. These morphological data suggested that PV$^+$ SC-PBGN and PV$^+$ SC-LPTN pathways may originate from different SC PV$^+$ neurons.

**SC PV$^+$ neurons encode threat-relevant visual signals.** Do PV$^+$ SC-PBGN and PV$^+$ SC-LPTN pathways transmit threat-relevant visual signals to the PBGN and LPTN? We injected AAV expressing double-floxed GCaMP6m into the SC of *PV-ires-Cre* mice to specifically label PV$^+$ neurons with GCaMP6m (Fig. 4a, b and Supplementary Fig. 5a). Fiber photometry was validated by the simultaneous recording of action potentials and calcium transients from GCaMP6m-expressing PV$^+$ neurons in the SuG and Op layers of acute SC slices (Fig. 4c).

In the in vivo experiments, we implanted an optic fiber above the medial SC, which monitors the upper visual field[26], of head-fixed anesthetized mice to record calcium transients from a group of GCaMP6m-expressing PV$^+$ neurons (Fig. 4d). A virtual soccer ball moving in controlled velocities and directions in the contralateral visual field was used as a visual stimulus (Fig. 4d). We used a soccer ball composed of black-and-white squares[41–43] instead of a plain dark disk as the looming visual stimulus because the black and white squares simultaneously expanded or shrank so that the ambient light intensity did not change and visual responses were not contaminated by a response component to ambient light change[44]. Although SC PV$^+$ neurons responded to the soccer ball moving in all six directions

(X+, X−, Y+, Y−, Z+, Z−), they had the strongest visual response to the ball moving on a collision course toward the eye (Z+; Supplementary Fig. 5b, c). The response onset time to the looming ball was defined as the time point when the Ca$^{2+}$ signal reaches 15% of response peak (Supplementary Fig. 5d). The response onset time of SC PV$^+$ neurons depended on the diameter (*D*) and velocity (*V*) of the ball (Fig. 4f−i) and was linearly correlated with the square root of the *D/V* ($R = 0.985$, $P = 0.006$, $n = 9$ mice; Fig. 4j). The response peak was close to the time to collision (Fig. 4f−i) and was independent of the square root of the *D/V* of the ball (Fig. 4k). All the receptive fields (9/9) of the SC PV$^+$ neurons recorded from nine mice were located above the temporal-nasal meridian (Supplementary Fig. 5e), suggesting that these visual neurons might monitor visual threats in the upper visual field.

To examine whether PV$^+$ SC-PBGN and PV$^+$ SC-LPTN pathways could transmit the threat-relevant visual signals from SC PV$^+$ neurons to the PBGN and LPTN, we recorded Ca$^{2+}$ responses from the GCaMP6m$^+$ axon terminals of SC PV$^+$ neurons in the LPTN and PBGN with fiber photometry (Fig. 4e). Smaller but robust visual responses to the ball moving on a collision course were recorded from the axon terminals in the PBGN and LPTN (Fig. 4l, m). These threat-relevant visual signals transmitted by PV$^+$ SC-PBGN and PV$^+$ SC-LPTN pathways showed mild but insignificant differences in response onset time and response peak time (Fig. 4n). These data indicate that both PBGN and LPTN receive threat-relevant visual signals from SC PV$^+$ neurons.

**Divergent pathways trigger dimorphic defensive behaviors.** We then asked how these threat-relevant visual signals from the SC to the PBGN and LPTN trigger dimorphic defensive behaviors. AAV-DIO-ChR2-mCherry was injected into the SC of *PV-ires-Cre* mice (Supplementary Fig. 6a), resulting in the specific expression of ChR2-mCherry in SC PV$^+$ neurons[45] (Fig. 5a and Supplementary Fig. 6b). In acute SC slices, a light-pulse train (473 nm, 1 ms, 10 Hz, 20 mW) reliably triggered action potentials from ChR2-mCherry-expressing SC PV$^+$ neurons in the SuG and Op layers (Fig. 5b). In acute PBGN or LPTN slices treated with TTX and 4-AP (Supplementary Fig. 6c), isolated light pulses (473 nm, 1 ms, 20 mW) evoked robust PSCs that were predominantly blocked by perfusion of D-AP5 and CNQX (Supplementary Fig. 6d, e). These data indicate that neurons in the PBGN and LPTN receive monosynaptic innervations from SC PV$^+$ neurons.

To specifically activate the PV$^+$ SC-PBGN and PV$^+$ SC-LPTN pathways, we implanted an optic fiber above the ChR2-mCherry$^+$ axon terminals in the PBGN and LPTN (Fig. 5c). Optogenetic activation (473 nm, 20 ms, 10 Hz, 5 s, 20 mW) of the PV$^+$ SC-PBGN pathway triggered impulsive escape (latency = 109 ± 19 ms; duration = 1.1 ± 0.5 s) followed by long-lasting freezing (duration = 47 ± 12 s), whereas activation of the PV$^+$ SC-LPTN pathway induced immediate freezing (latency = 126 ± 21 ms; duration = 8.6 ± 3.1 s; Video 4; Fig. 5d, e). As a control experiment, optogenetic activation of the PV$^+$ SC-LDTN pathway did not elicit obvious defensive behaviors in mice (Supplementary Fig. 6f, g). For each mouse, we measured its locomotion speed before, during, and after optogenetic activation of the PV$^+$ SC-PBGN (Fig. 5f) and PV$^+$ SC-LPTN pathways (Fig. 5i). The distribution of LSI$_{during\ stimulation}$ and LSI$_{after\ stimulation}$ indicated that activation of PV$^+$ SC-PBGN triggered Type I defensive behavior (Fig. 5g), with a dramatic speed increase during light stimulation and decrease after light stimulation (Fig. 5h). In contrast, activation of the PV$^+$ SC-LPTN pathway induced Type II defensive behavior (Fig. 5j), with

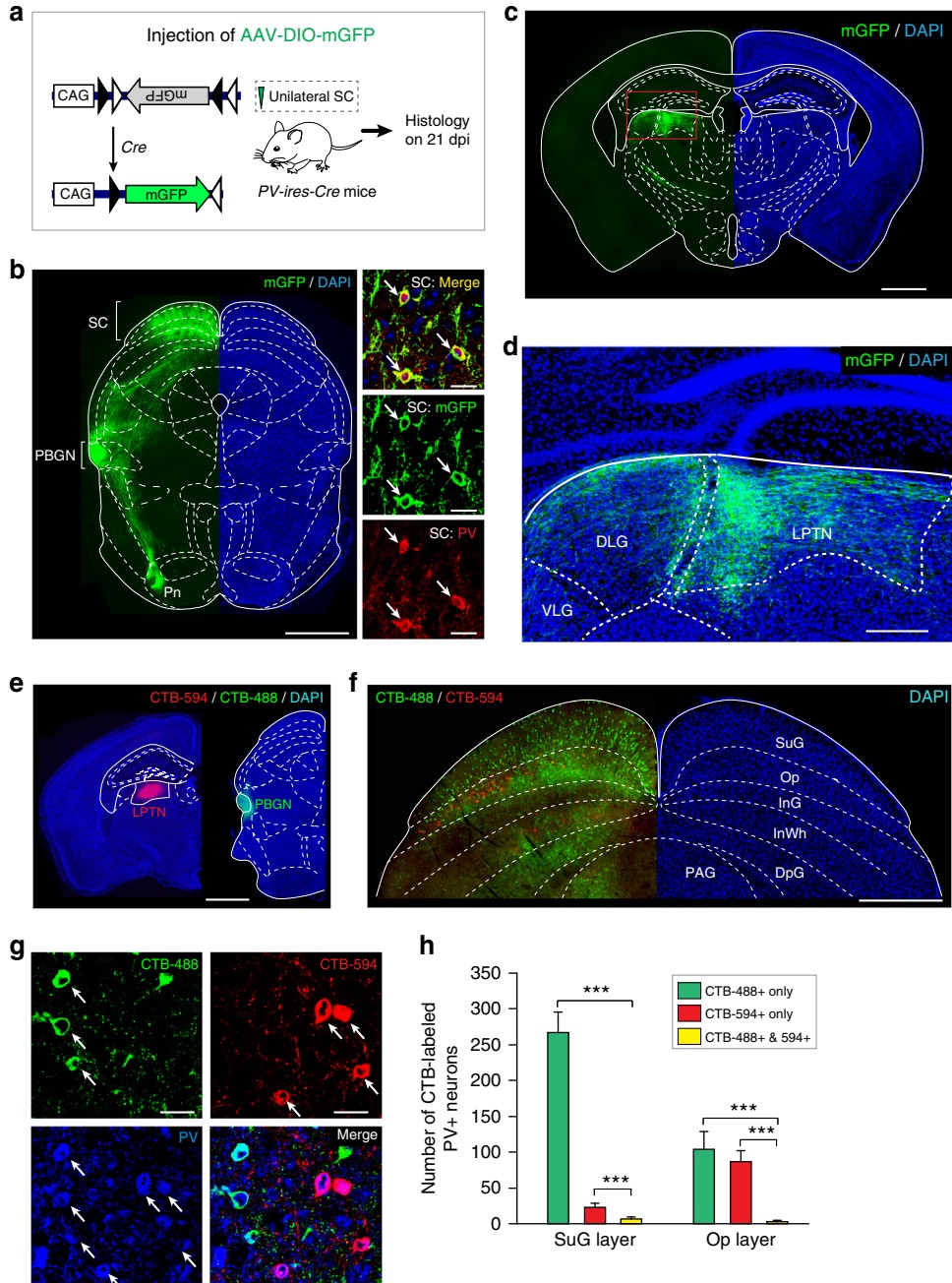

**Fig. 3** Distinct SC PV$^+$ neurons divergently project to the PBGN and LPTN. **a** Schematic diagram showing AAV-DIO-mGFP injection into the SC of *PV-ires-Cre* mice. **b** Coronal section (left) and example micrographs (right) showing specific expression of mGFP in SC PV$^+$ neurons. Arrows indicate PV$^+$ neurons. Scale bars, 1 mm (left) and 30 μm (right). For quantitative analyses, see Supplementary Fig. 4a. **c, d** Coronal section (**c**) and its cropped region (**d**) showing mGFP$^+$ axon terminals of SC PV$^+$ neurons in the LPTN and other adjacent thalamic nuclei. For the whole-brain mapping of SC PV$^+$ neuron targets, see Supplementary Fig. 4a and 4b. Scale bars, 1 mm (**c**) and 200 μm (**d**). **e** Schematic diagram showing ipsilateral injection of CTB-594 into the LPTN and CTB-488 into the PBGN of the same mice. Scale bar, 1 mm. **f** Example coronal section of the ipsilateral SC showing the distribution of retrogradely labeled cells projecting to the PBGN (CTB-488$^+$, green) and LPTN (CTB-594$^+$, red). Scale bar, 0.5 mm. For single-channel micrographs, see Supplementary Fig. 4c. **g** Example micrographs from Op layer showing a large proportion of PBGN-projecting (CTB-488$^+$) and LPTN-projecting neurons (CTB-594$^+$) were positive for PV. Arrows indicate CTB-488$^+$ and CTB-594$^+$ neurons that are also positive for PV. Scale bars, 25 μm. **h** Quantitative analyses of the laminar distribution of SC PV$^+$ neurons projecting to the PBGN (CTB-488$^+$) and LPTN (CTB-594$^+$). Data in (**h**) are means ± SEM (error bars, $n = 3$ mice). Statistical analysis was Student's $t$ test (***$P < 0.001$)

a significant decrease in locomotion speed during and after light stimulation (Fig. 5k). Thus, the behavioral patterns induced by optogenetic activation of the PV$^+$ SC-PBGN and PV$^+$ SC-LPTN pathways roughly mimicked the dimorphic defensive behaviors triggered by looming visual stimuli.

**PBGN and LPTN neurons respond to looming visual stimuli.** Our data, at this point, indicated that SC PV$^+$ neurons, as a key neuronal subtype in the SC, send threat-relevant visual signals divergently to the PBGN and LPTN to trigger Type I and Type II defensive behaviors, respectively. How the PBGN and LPTN

neurons adopt these threat-relevant signals to mediate these dimorphic defensive behaviors remained unclear.

To address this question, we first examined the cell types in the PBGN and LPTN. By immunostaining of neuronal marker NeuN together with glutamate or GABA, we found that neurons in the PBGN and LPTN were mostly, if not exclusively, glutamatergic (Supplementary Fig. 7a, b). The glutamatergic neurons in the

PBGN and LPTN could be reliably labeled by injecting small volumes of AAV expressing double-floxed molecular tools (EGFP: Fig. 6a, b and Supplementary Fig. 7c, d; GCaMP6m: Fig. 6c, d and Supplementary Fig. 7e, f; ChR2-mCherry: Fig. 7a, b and Supplementary Fig. 7g, h; EGFP-2A-TeNT: Fig. 8a, b and Supplementary Fig. 7i, j) into the PBGN (200 nl) and LPTN (400 nl) of *vGlut2-ires-Cre* mice with a nano-liter injector.

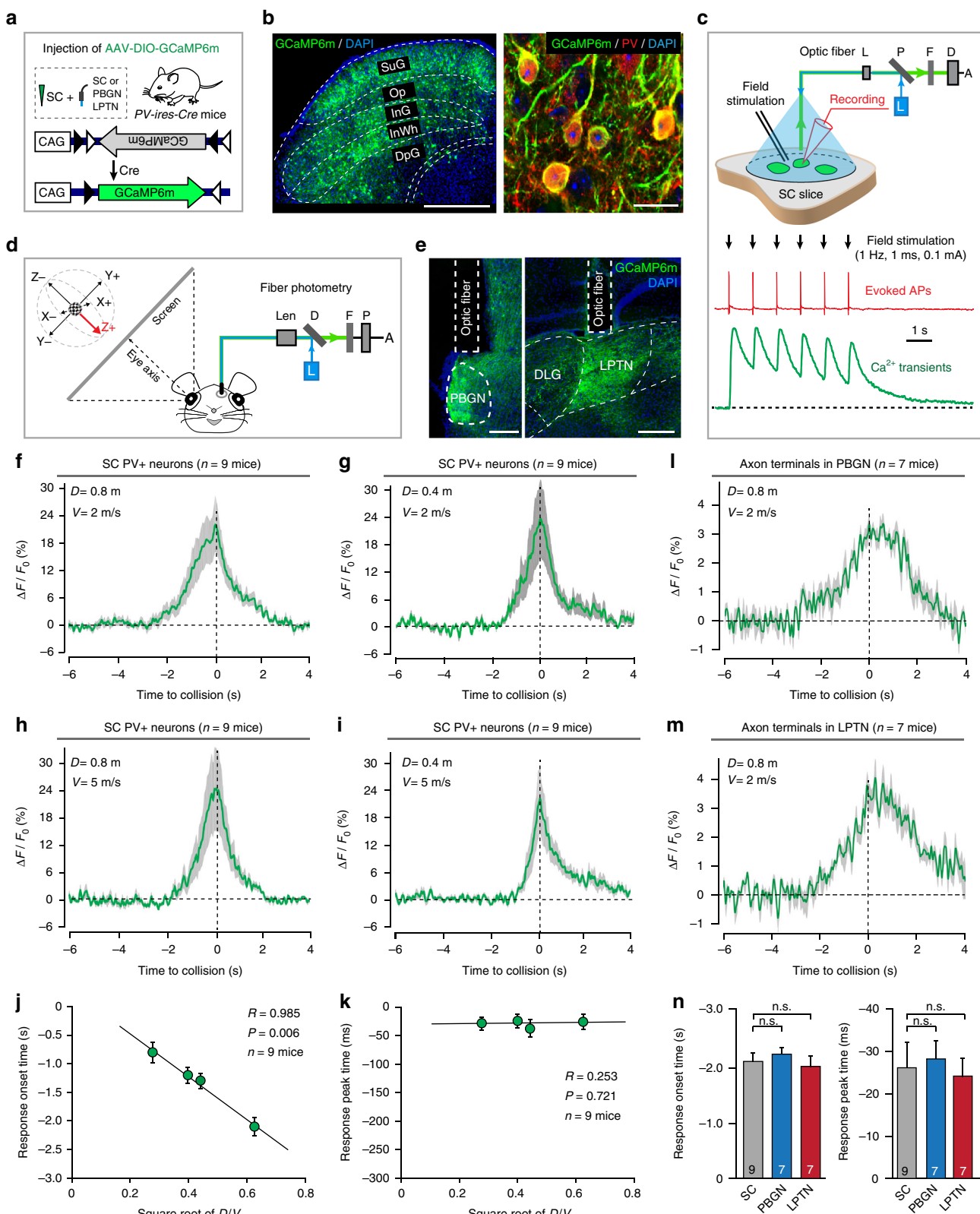

To determine whether glutamatergic neurons in the PBGN and LPTN receive looming visual signals, we recorded their activity in freely moving mice. A small volume of AAV-DIO-GCaMP6m was injected into the PBGN (200 nl) and LPTN (400 nl) of *vGlut2-ires-Cre* mice, resulting in localized and specific expression of GCaMP6m in glutamatergic neurons (Fig. 6c, d and Supplementary Fig. 7e, f). The calcium transients from these neurons were recorded with an optic fiber implanted above the PBGN or LPTN (Fig. 6c, d) while freely moving mice were subjected to looming visual stimuli in the arena (Fig. 6e). The mice exploring in the arena were constantly changing the body position and head orientation. To minimize the variation in visual responses caused by these factors, we only provided one cycle of looming stimulation. The looming visual stimulation that triggered mouse dimorphic defensive behaviors evoked robust calcium transients from glutamatergic neurons in the PBGN (Video 5; Fig. 6f; $n = 6$ mice) and LPTN (Video 6; Fig. 6h; $n = 6$ mice). The PBGN neurons exhibited calcium transients with similar amplitude and kinetic properties regardless of Type I or Type II behavioral patterns (Fig. 6g). Likewise, calcium responses of the LPTN neurons did not show selectivity for the two behavioral patterns (Fig. 6i). These data indicate that, in freely moving mice, PBGN and LPTN neurons are co-activated during visually triggered dimorphic defensive behaviors.

**PBGN and LPTN neurons trigger dimorphic defensive behaviors.** To examine the behavioral relevance of the activity of PBGN and LPTN neurons in response to looming visual stimulation, we injected small volumes of AAV-DIO-ChR2-mCherry into the PBGN (200 nl) and LPTN (400 nl) of *vGlut2-ires-Cre* mice, followed by optic fiber implantation above these nuclei (Fig. 7a, b). We observed localized and specific expression of ChR2-mCherry in glutamatergic PBGN and LPTN neurons (Fig. 7a, b and Supplementary Fig. 7g, h). In acute PBGN and LPTN slices, the light-pulse train reliably triggered spikes from ChR2-mCherry[+] PBGN and LPTN neurons (Fig. 7c). Furthermore, the light-evoked PSCs from adjacent non-infected PBGN and LPTN neurons were almost completely eliminated by perfusion of D-AP5 and CNQX (Fig. 7d−f), indicating again that ChR2-mCherry was exclusively targeted to glutamatergic PBGN and LPTN neurons.

Optogenetic activation of PBGN neurons (473 nm, 20 ms, 10 Hz, 5 s, 20 mW) triggered immediate escape (latency = 117 ± 28 ms; duration = 0.9 ± 0.4 s) followed by freezing (duration = 29 ± 7 s), whereas activation of LPTN neurons induced immediate freezing (latency = 113 ± 21 ms; duration = 14 ± 6.2 s; Video 7). For each mouse, we measured the locomotion speed before, during, and after optogenetic activation of PBGN (Fig. 7g) and LPTN neurons (Fig. 7j). Analyses of LSI$_{during\ stimuli}$ and LSI$_{after\ stimuli}$ indicated that activation of the PBGN neurons triggered Type I defensive behavior (Fig. 7h), with a dramatic speed increase during stimuli and decrease after stimuli (Fig. 7i). In contrast, activation of the

LPTN neurons induced Type II defensive behavior (Fig. 7k), with a significant decrease in locomotion speed both during and after stimuli (Fig. 7l). Thus, the behavioral patterns induced by optogenetic activation of PBGN and LPTN neurons, similar to those induced by activation of the PV[+] SC-PBGN and PV[+] SC-LPTN pathways, mimicked the dimorphic defensive behavioral patterns triggered by looming visual stimuli.

**Effects of selective silencing of LPTN or PBGN neurons.** Up to this point, our data indicated that the PBGN and LPTN utilize threat-relevant visual signals from the SC PV[+] neurons to trigger Type I and Type II defensive behavioral patterns, respectively. However, it was unclear how these two nuclei coordinately control the generation of dimorphic defensive behaviors. The co-activation of PBGN and LPTN neurons by looming visual stimulation in freely moving mice (Fig. 6f–i) raised the possibility that they might work in competition for the dominance of behavioral outcome. If this hypothesis is correct, bilateral inactivation of either nucleus would result in mouse behavior dominated by the other nucleus.

We tested this hypothesis by selectively silencing the bilateral PBGN or LPTN with TeNT. Before examining the behavioral effects of PBGN or LPTN inactivation, we first measured the efficiency of TeNT-induced synaptic inactivation. A mixture of AAV-DIO-ChR2-mCherry and AAV-DIO-EGFP-2A-TeNT was injected into the unilateral PBGN or LPTN of *vGlut2-ires-Cre* mice, resulting in the co-expression of EGFP and ChR2-mCherry in glutamate[+] neurons (Supplementary Fig. 8a, d). The amplitude of light-evoked PSCs from adjacent ChR2-mCherry-negative cells in acute PBGN and LPTN slices was strongly reduced by TeNT expression in glutamatergic neurons (Supplementary Fig. 8b, c, e, f), suggesting that TeNT efficiently inactivated PBGN and LPTN glutamatergic neurons.

To examine the effects of PBGN and LPTN inactivation on visually triggered dimorphic defensive behaviors, we injected AAV-DIO-EGFP-2A-TeNT into the bilateral PBGN (200 nl) and LPTN (400 nl) of *vGlut2-ires-Cre* mice with a nano-liter injector, resulting in EGFP and TeNT expression in glutamatergic PBGN and LPTN neurons (Fig. 8a, b and Supplementary Fig. 7i, j). We measured locomotion speed before, during, and after looming visual stimuli of mice with selective inactivation of PBGN (Fig. 8c) and LPTN neurons (Fig. 8f). Strikingly, the distribution of LSI$_{during\ stimuli}$ and LSI$_{after\ stimuli}$ indicated that synaptic inactivation of the bilateral PBGN resulted in the disappearance of the Type I defensive behavioral pattern and dominance of the Type II pattern (Fig. 8d; Video 8). The Type II pattern exhibited by mice with bilateral PBGN inactivation slightly differed from that of control mice, with a significant increase in locomotion speed during freezing phase after visual stimuli (Fig. 8e). Conversely, all mice with bilateral LPTN inactivation showed the Type I behavioral pattern (Fig. 8g; Video 9), with locomotion speed similar to that of control mice with the Type I pattern

**Fig. 4** SC PV[+] neurons transmit threat-relevant visual signals to the PBGN and LPTN. **a** Diagram showing AAV-DIO-GCaMP6m injection into the SC followed by optic fiber implantation above the SC, PBGN or LPTN of *PV-ires-Cre* mice. **b** Micrographs of the SC showing expression of GCaMP6m in PV[+] neurons. Scale bars, 0.5 mm (left) and 25 μm (right). For single-channel micrographs and quantitative analyses, see Supplementary Fig. 5a. **c** Validation of fiber photometry by simultaneous recording of action potentials (APs) and Ca[2+] transients from SC PV[+] neurons in SC slices. The evoked APs (red) and Ca[2+] transients (green) were synchronized with field stimulation train. **d** Schematic diagram showing in vivo fiber photometry to record Ca[2+] transients in anesthetized mice. A virtual soccer ball moving in six orthogonal directions (±X, ±Y, ±Z) was displayed on an oblique screen facing directly to the eye axis. **e** Fiber tracks above the GCaMP6[+] axon terminals in the PBGN (left) and LPTN (right). Scale bars, 0.2 mm. **f−i** Averaged peri-stimulus time course of Ca[2+] transients from SC PV[+] neurons of nine mice to the looming ball with controlled velocity ($V = 2$ or $5\ m\ s^{-1}$) and diameter ($D = 0.4$ or $0.8\ m$). For responses in other directions, see Supplementary Fig. 5b. **j, k** Correlation analyses of response onset time (**j**) or peak time (**k**) vs. square root of D/V of the looming ball. **l, m** Time course of Ca[2+] transients from PV[+] axon terminals in the PBGN (**l**) and LPTN (**m**) to the looming ball ($D = 0.8\ m$, $V = 2\ m\ s^{-1}$). **n** Comparison of response onset time (left) and peak time (right) between SC PV[+] neurons and their axon terminals in the PBGN or LPTN. Cloudy area in (**f−i, l, m**) indicates SEM of the averaged data. Data in (**j, k, n**) are means ± SEM (error bars). Statistical analysis was Student's $t$ test (n.s. $P > 0.1$)

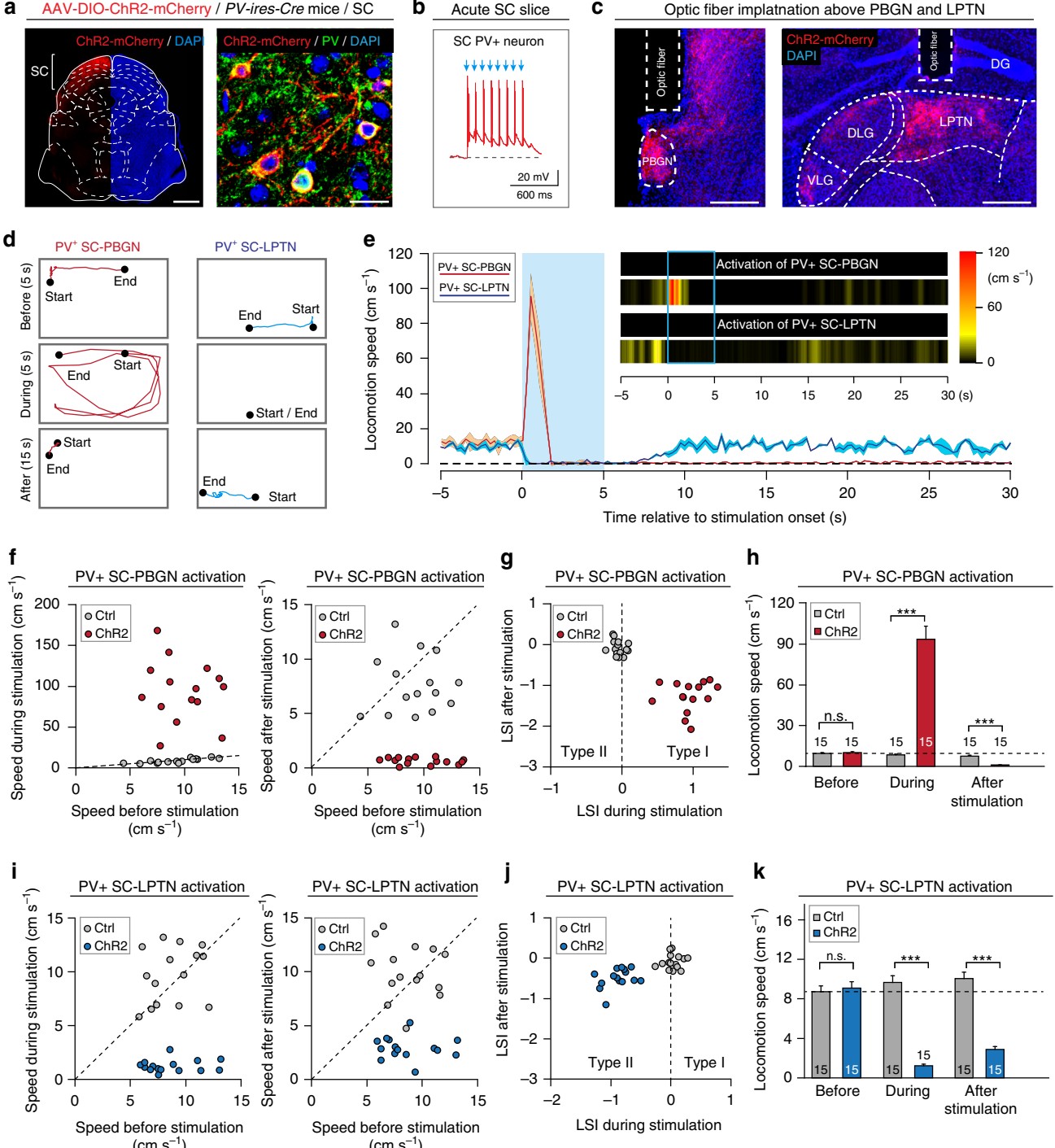

**Fig. 5** Activation of PV+ SC-LPTN pathway and PV+ SC-PBGN pathway mimicked dimorphic defensive behavioral patterns. **a** Coronal section (left) and micrograph (right) showing specific expression of ChR2-mCherry in SC PV+ neurons after AAV-DIO-ChR2-mCherry injection. Scale bars, 1 mm (left) and 20 μm (right). For single-channel micrographs and quantitative analyses, see Supplementary Fig. 6b. **b** A light-pulse train (473 nm, 10 Hz, 1 ms, 8 pulses, 20 mW) reliably triggered APs from ChR2-mCherry+ neurons in the acute SC slices. **c** Example micrographs showing optic fiber tracks above the ChR2-mCherry+ axon terminals in the PBGN (left) and LPTN (right). Scale bars, 0.3 mm. **d** Locomotion trails of example mice in home-cages before, during, and after optogenetic activation (10 Hz, 20 ms, 5 s, 20 mW) of PV+ SC-PBGN and PV+ SC-LPTN pathways. **e** Averaged time course of locomotion speed before, during, and after activation of the PV+ SC-PBGN (red, n = 15 mice) and PV+ SC-LPTN (blue, n = 15 mice) pathways. Cloudy area indicates SEM of the averaged data. Inset, heat-maps of locomotion speed of example mice. **f**, **i** Distributions of locomotion speed during vs. before (left) and after vs. before (right) optogenetic activation of the PV+ SC-PBGN (**f**) and PV+ SC-LPTN pathways (**i**). **g**, **j** Distribution of LSI$_{during\ stimulation}$ and LSI$_{after\ stimulation}$ of mice with (ChR2) or without (Ctrl) activation of the PV+ SC-PBGN (**g**) or SC-LPTN (**j**) pathways. **h**, **k** Quantitative analyses of locomotion speed before, during, and after optogenetic activation of the PV+ SC-PBGN (**h**) and SC-LPTN (**k**) pathways. For the analyses of PV+ SC-LDTN pathway, see Supplementary Fig. 6f and g. Data in (**h**, **k**) are means ± SEM (error bars). Numbers of mice are indicated in the bars. Statistical analysis was Student's t test (***P < 0.001; n.s. P > 0.1)

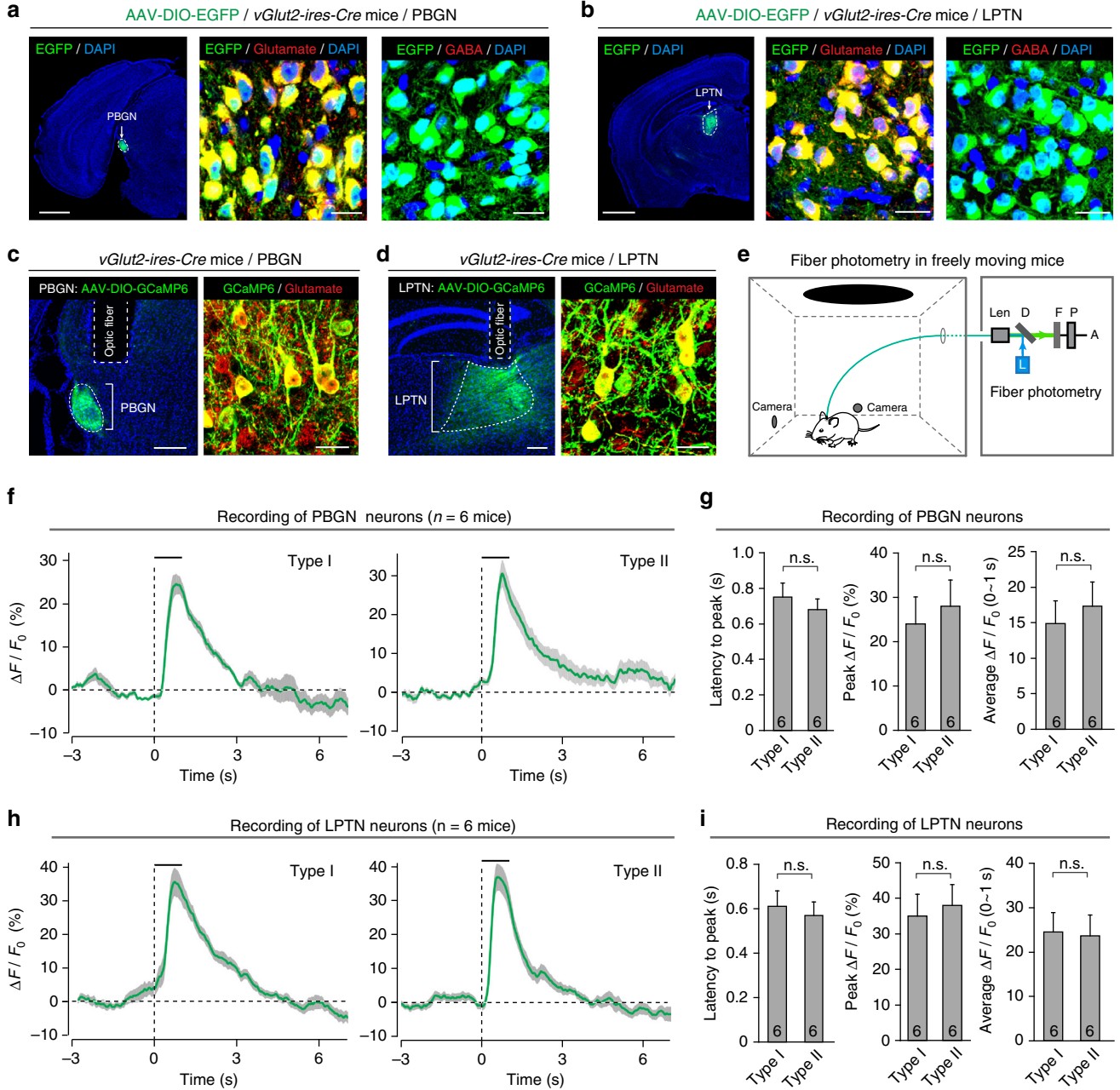

**Fig. 6** Glutamatergic neurons in the PBGN and LPTN respond to looming visual stimuli. **a**, **b** AAV-DIO-EGFP was injected into the PBGN (**a**, left) and LPTN (**b**, left) of *vGlut2-ires-Cre* mice, resulting in EGFP expression in glutamate$^+$ neurons in the PBGN (**a**, middle) and LPTN (**b**, middle). Note there were almost no GABA$^+$ neurons in the PBGN (**a**, right) or LPTN (**b**, right). Scale bars, 1 mm (left) and 25 μm (middle, right) for (**a**) and (**b**). **c**, **d** Example coronal sections with optic fiber tracks above the PBGN (**c**, left) and LPTN (**d**, left), and micrographs showing specific expression of GCaMP6 in glutamate$^+$ neurons in the PBGN (**c**, right) and LPTN (**d**, right). Scale bars, 0.2 mm (left) and 30 μm for both (**c**) and (**d**). For single-channel micrographs and quantitative analyses, see Supplementary Fig. 7c−f. **e** Schematic diagram showing recording of Ca$^{2+}$ transients from freely moving mice in response to looming visual stimulation. **f**, **h** Averaged peri-stimulus time course of Ca$^{2+}$ transients from PBGN (**f**, $n = 6$ mice) and LPTN neurons (**h**, $n = 6$ mice) of freely moving mice exhibiting Type I (left) and Type II (right) defensive behavioral patterns in response to one cycle of looming stimulation (horizontal bar). Cloudy area indicates SEM of the averaged data. **g**, **i** Quantitative analyses of latency to peak (left), peak $\Delta F/F_0$ (middle), and average $\Delta F/F_0$ of calcium transients (right) during visual stimulation from PBGN (**g**) and LPTN neurons (**i**). Data in (**g**, **i**) are means ± SEM (error bars). Numbers of mice are indicated in the bars. Statistical analysis was Student's *t* test (n.s. $P > 0.1$)

(Fig. 8h). These data suggest that the PBGN and LPTN mutually compete for the dominance of behavioral outcome of Type I and II patterns in response to looming visual stimuli.

**Effects of selective exciting PBGN and LPTN neurons.** To further examine how the PBGN and LPTN coordinately control the generation of dimorphic defensive behaviors, we selectively

increased the intrinsic excitability of PBGN or LPTN neurons (Fig. 9). NaChBac, a bacterial voltage-dependent sodium channel, has an activation threshold more negative than that of native sodium channel in neuron[46]. This property has made it a tool to manipulate neuronal excitability in mice[47]. We bilaterally injected AAV-DIO-EGFP-2A-NaChBac into the PBGN or LPTN of *vGlut2-ires-Cre* mice, resulting in expression of EGFP in PBGN

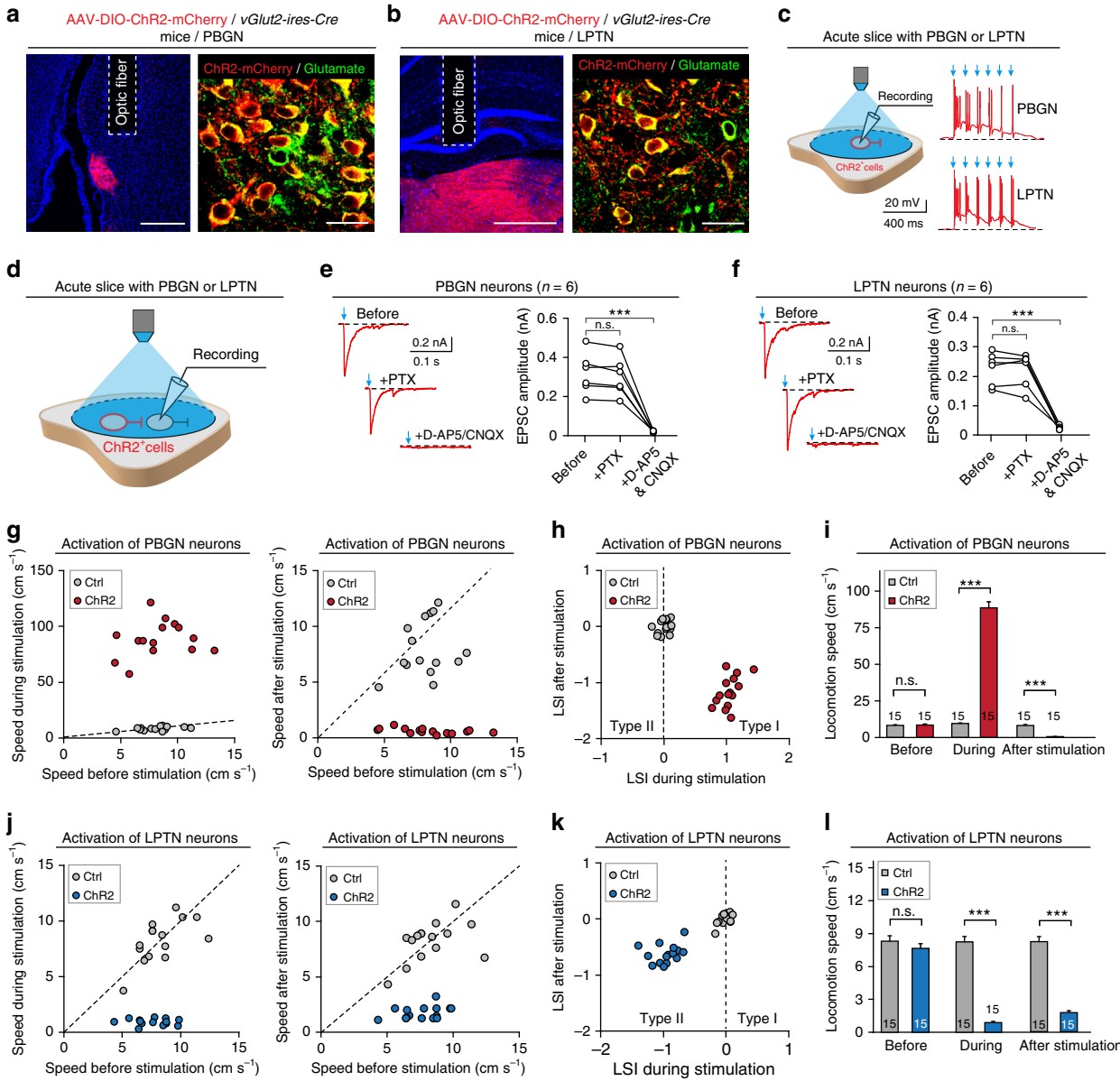

**Fig. 7** Selective activation of glutamatergic neurons in the PBGN and LPTN triggers dimorphic defensive response patterns. **a**, **b** AAV-DIO-ChR2-mCherry was locally injected into the PBGN (**a**, left) and LPTN (**b**, left) of *vGlut2-ires-Cre* mice, resulting in the specific expression of ChR2-mCherry in glutamate[+] neurons in the PBGN (**a**, right) and LPTN (**b**, right). Scale bars, 0.4 mm (left) and 30 μm (right) for (**a**) and (**b**). For single-channel micrographs and quantitative analyses, see Supplementary Fig. 7g and h. **c** In acute slice with the PBGN or LPTN, light-pulse trains (473 nm, 1 ms, 10 Hz, 6 pulses, 20 mW) reliably evoked APs from ChR2-mCherry[+] neurons in the PBGN and LPTN. **d** Schematic diagram showing recording of light-evoked PSCs from ChR2-mCherry-negative neurons in acute slice with PBGN or LPTN. **e**, **f** Effects of PTX and D-AP5/CNQX on light-evoked PSCs recorded from ChR2-mCherry-negative neurons in the PBGN (**e**) and LPTN (**f**). **g**, **j** Distributions of locomotion speed of mice during vs. before (left) and after vs. before (right) optogenetic activation of PBGN (**g**) and LPTN neurons (**j**). **h**, **k** Distribution of LSI$_{during\ stimulation}$ and LSI$_{after\ stimulation}$ of mice with (ChR2) or without (Ctrl) optogenetic activation of PBGN (**h**) and LPTN neurons (**k**). **i**, **l** Quantitative analyses of speed before, during, and after optogenetic activation of PBGN (**i**) and LPTN neurons (**l**). Data in (**i**, **l**) are means ± SEM (error bars). Numbers of mice are indicated in the bars. Statistical analysis (**e**, **f**, **i**, **l**) was Student's *t* test (***$P < 0.001$; n.s. $P > 0.1$)

and LPTN neurons (Fig. 9a, b). The efficiency of AAV-DIO-EGFP-2A-NaChBac to increase neuronal excitability was examined by whole-cell recording from the EGFP[+] PBGN or LPTN neurons in acute slices, with AAV-DIO-EGFP as a control (Fig. 9c). The EGFP[+] neurons in both PBGN and LPTN slices infected by AAV-DIO-EGFP-2A-NaChBac (NaChBac) fired more action potentials in response to depolarizing currents than the control neurons (Ctrl) (one-way ANOVA, PBGN: $P < 0.001$, LPTN: $P < 0.001$; Fig. 9d–g).

Interestingly, selective increase of excitability of PBGN neurons by NaChBac expression increased the proportion of mice with Type I behavioral pattern (From 66.7 to 100%; Fig. 9h−j). NaChBac expression in bilateral LPTN increased the proportion of mice with Type II behavioral pattern (From 26.7 to 60%; Fig. 9k−m). These data indicated that increased excitability of PBGN or LPTN neurons can shift the behavioral outcome to the pattern dominated by PBGN or LPTN, respectively. Together, the data of synaptic inactivation (Fig. 8) and of increased excitability (Fig. 9) suggested that the SC-PBGN and SC-LPTN pathways form two

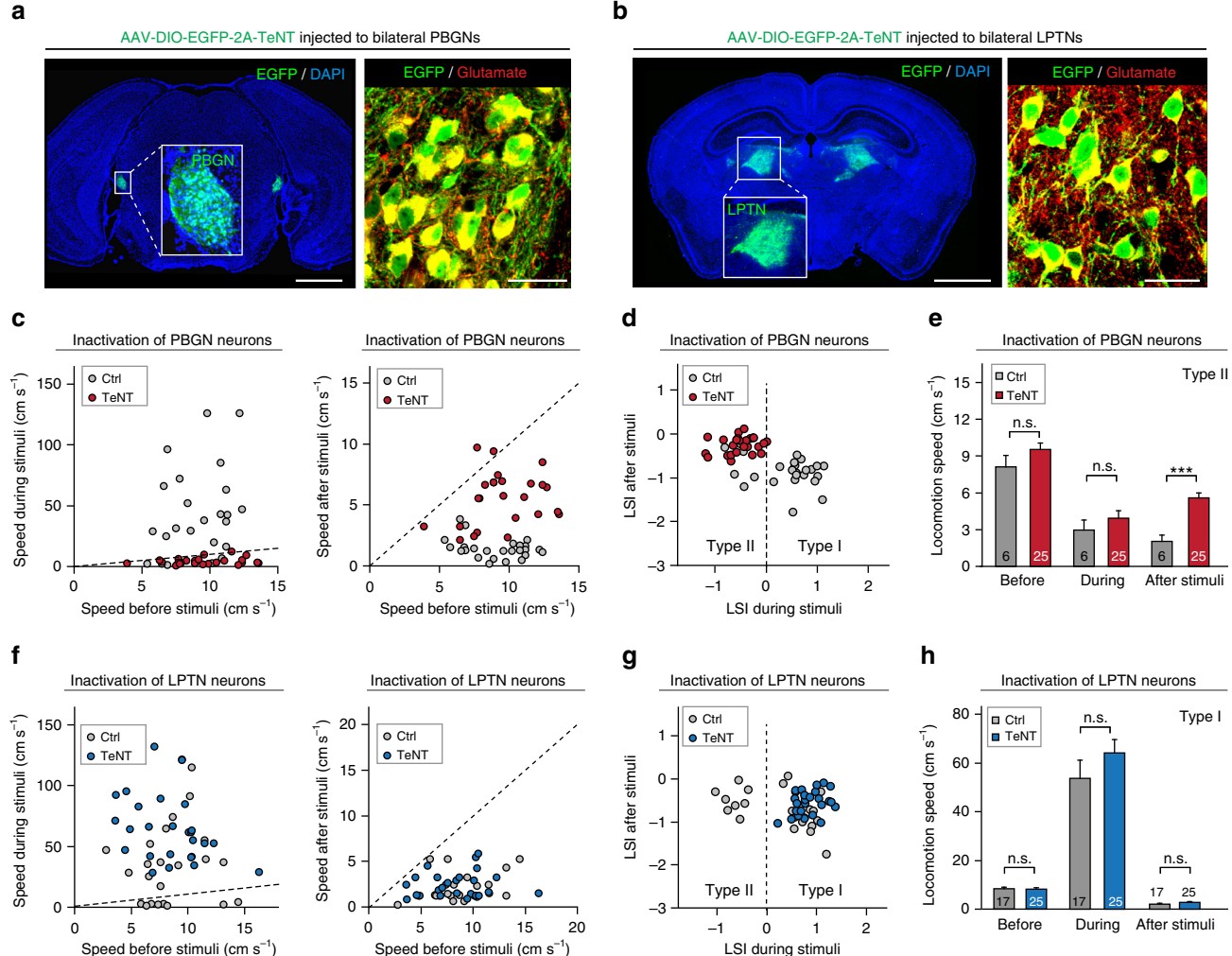

**Fig. 8** Effects of selective inactivation of bilateral PBGN or LPTN on dimorphic defensive behaviors. **a**, **b** AAV-DIO-EGFP-2A-TeNT was locally injected into the bilateral PBGN (**a**, left) and LPTN (**b**, left) of *vGlut2-ires-Cre* mice, resulting in the specific expression of EGFP in glutamate[+] neurons in the PBGN (**a**, right) and LPTN (**b**, right). Scale bars, 1.5 mm (left) and 30 μm (right) for (**a**) and (**b**). For single-channel micrographs and quantitative analyses, see Supplementary Fig. 7i and j. For validation of TeNT effect on neurotransmitter release, see Supplementary Fig. 8. **c**, **f** Distribution of locomotion speed during vs. before (left) and after vs. before (right) looming visual stimuli of mice with (TeNT) or without (Ctrl) inactivation of PBGN (**c**) and LPTN (**f**) neurons. **d**, **g** Distribution of LSI$_{during stimuli}$ and LSI$_{after stimuli}$ of mice with (TeNT) or without (Ctrl) selective inactivation of PBGN (**d**) and LPTN (**g**) neurons. **e** Quantitative analyses of the effects of PBGN inactivation on locomotion speed of mice exhibiting the Type II defensive behavioral pattern. **h** Quantitative analyses of the effect of LPTN inactivation on locomotion speed of mice showing the Type I defensive behavioral pattern. Data in (**e**, **h**) are means ± SEM (error bars). Numbers of mice are indicated in the graphs. Statistical analysis was Student's *t* test (***$P < 0.001$; n.s. $P > 0.1$)

mutually competing circuit modules to determine the outcome of dimorphic defensive behavioral patterns in mice. This conclusion was illustrated in the summarized diagrams (Fig. 10).

## Discussion

Threat-relevant sensory stimuli trigger either escape or freezing behaviors, yet the key neural circuits to generate these dimorphic defensive behaviors in the mammalian brain remain unclear. We found that the dimorphic defensive behavioral patterns in mice triggered by looming visual stimuli (Fig. 1) were mediated by SC PV[+] neurons, a key neuronal subtype to trigger defensive behaviors (Fig. 2). Two distinct groups of PV[+] neurons formed divergent visual pathways to the PBGN and LPTN, respectively (Fig. 3). These pathways transmitted threat-relevant visual signals to the PBGN and LPTN in both anesthetized (Fig. 4) and freely moving mice (Fig. 6). Selective activation of the PV[+] SC-PBGN and PV[+] SC-LPTN pathways (Fig. 5), or activation of the glutamatergic PBGN and LPTN neurons (Figs. 7, 9), mimicked these

dimorphic behavioral patterns. Bilateral silencing of the PBGN or LPTN resulted in dominance of the behavioral pattern controlled by the other nucleus (Fig. 8), suggesting that the PBGN and LPTN mutually compete for behavioral outcome. Together, these data suggest a dual-circuit winner-take-all mechanism that might be used by the SC to orchestrate dimorphic defensive behaviors in mammals.

Freezing and fleeing are major forms of defensive behavior across species[3,5,7,9]. Three general factors determining defensive behavioral patterns in response to threats have been proposed[4]: (1) environmental context (e.g., access to shelter); (2) threat stimulus features; and (3) individual differences (e.g., age and baseline hormone level). Regarding environmental context, a shelter in a behavioral arena can promote escape behavior[3]. For stimulus features, a small dark disk moving overhead simulating a distant cruising predator is more effective than looming visual stimulus[7] to induce freezing behavior[7]. For individual differences, it was observed that mice from different mouse colonies tend to

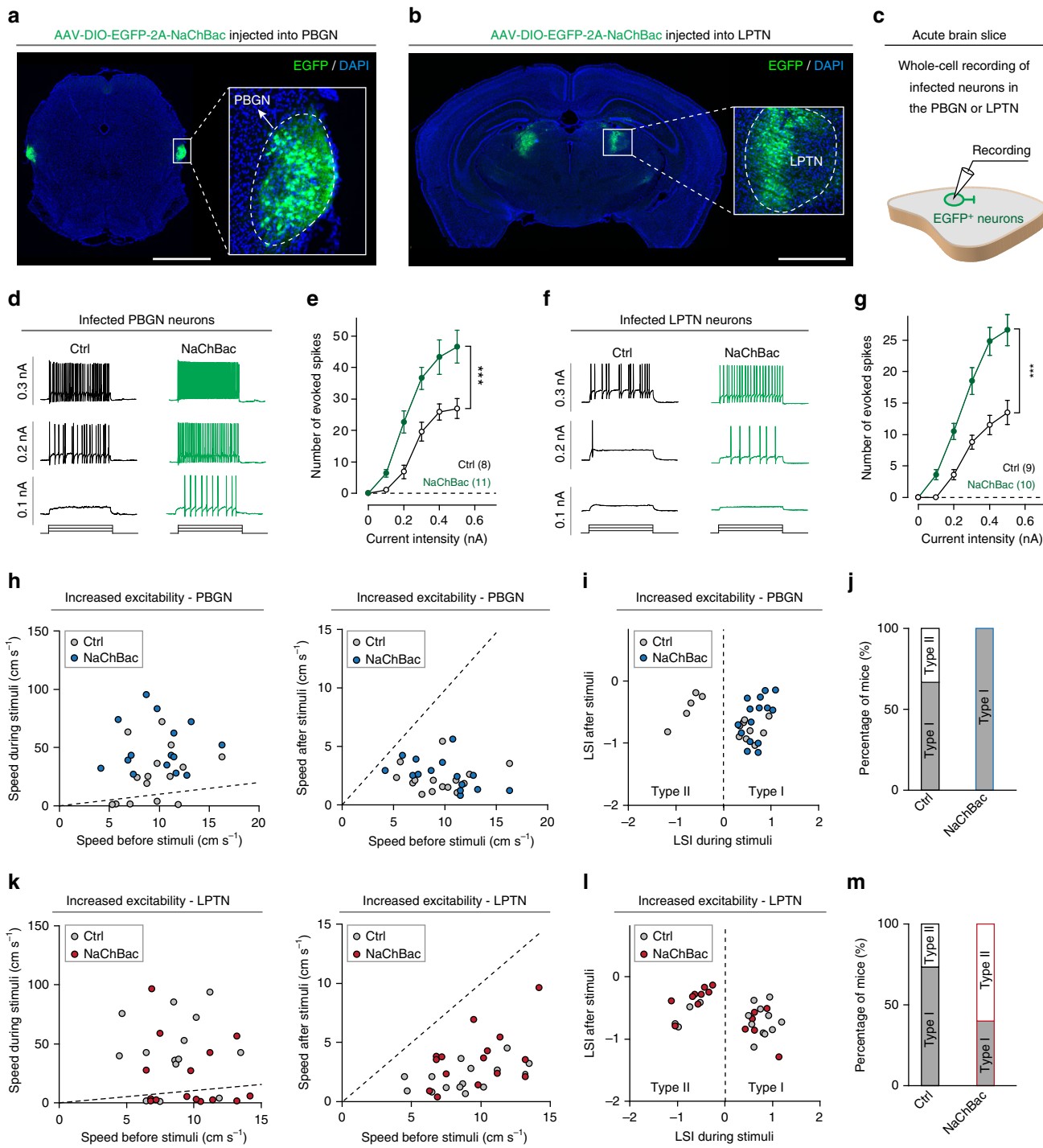

**Fig. 9** Effects of selective increase of intrinsic excitability of PBGN neurons or LPTN neurons on visually triggered dimorphic defensive behaviors. **a**, **b** AAV-DIO-EGFP-2A-NaChBac was locally injected into the bilateral PBGN (**a**) and LPTN (**b**) of *vGlut2-ires-Cre* mice, resulting in expression of EGFP in PBGN neurons (**a**, inset) and LPTN neurons (**b**, inset). Scale bars, 1.5 mm (**a**) and 2 mm (**b**). **c** Schematic diagram showing whole-cell recording of EGFP+ PBGN or LPTN neurons in acute slices. **d**, **e** Example traces (**d**) and quantitative analyses (**e**) of depolarization-induced spiking activity from EGFP+ PBGN neurons infected by AAV-DIO-EGFP (Ctrl) or AAV-DIO-EGFP-2A-NaChBac (NaChBac). **f**, **g** Example traces (**f**) and quantitative analyses (**g**) of depolarization-induced spiking activity from EGFP+ LPTN neurons infected by AAV-DIO-EGFP (Ctrl) or AAV-DIO-EGFP-2A-NaChBac (NaChBac). **h**, **k** Distribution of locomotion speed during vs. before (left) and after vs. before (right) looming visual stimuli of mice with (NaChBac) or without (Ctrl) increased excitability of PBGN neurons (**h**) and LPTN neurons (**k**). **i**, **l** Distribution of LSI$_{during\ stimuli}$ and LSI$_{after\ stimuli}$ of mice with (NaChBac) or without (Ctrl) selective increase of excitability in PBGN neurons (**i**) and LPTN neurons (**l**). **j**, **m** Quantitative analyses of the effects of increased excitability of PBGN neurons (**j**) or LPTN neurons (**m**) on the percentage of mice with Type I and Type II defensive behavioral patterns. Data in (**e**, **g**) are means ± SEM (error bars). Numbers of cells are indicated in the graphs. Statistical analysis was one-way ANOVA (***$P < 0.001$)

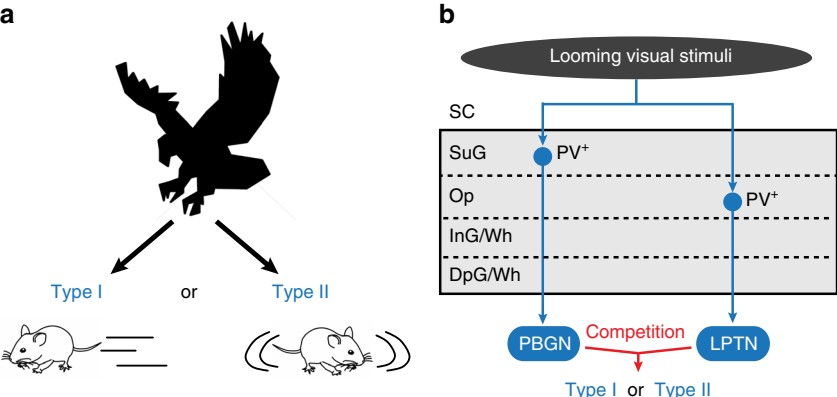

**Fig. 10** Summarized diagram illustrating the circuit mechanism underlying the action selection between dimorphic defensive behaviors. **a** Looming visual stimuli mimicking predator of mice trigger either Type I (escape-freezing) or Type II (freezing-only) defensive behavioral pattern. **b** Looming visual stimuli were encoded by PV$^+$ neurons in retinal-recipient layers (SuG and Op) of the SC. Distinct groups of SC PV$^+$ neurons transmit looming visual signals to the PBGN and LPTN. PBGN and LPTN then send these signals to the downstream targets to determine the behavioral outcome of mice, either Type I or Type II defensive behavioral pattern

show distinct behavioral patterns to the same looming visual stimuli[5].

Visual responses to looming stimuli have been a focus of studies. In the vertebrate brain, single-unit activity encoding looming visual stimuli was initially identified in rat SC[32], followed by characterization of looming responses in the SC of other mammalian species[34,42,48,49] and the optic tectum of non-mammals[43,50–52]. The looming detection might originate in the retina, because some retinal ganglion cells are sensitive to approaching visual stimuli[53]. The responses to looming visual stimuli in the SC might be mediated by excitation and inhibition in recurrent networks in the SC[50]. The cortical top-down innervations on SC neurons might enhance looming stimuli-induced visual responses[49].

In addition to retinal inputs, the mouse SC receives abundant afferent projections from the basal ganglia and cortical regions[54]. The afferents from the basal ganglia play a crucial role in rapid orienting behaviors such as saccadic eye movements[55,56]. Top-down cortical inputs from the primary visual cortex to the SC trigger "temporary arrest behavior" in mice[57]. It would be interesting to explore how these non-retinal inputs modulate the visually triggered defensive behaviors in future study.

The finding that both LPTN and PBGN are downstream targets of SC PV$^+$ neurons may have solved an apparent discrepancy between two recent studies[34,40]. One study[40] showed that visually triggered innate fear is processed by the SC-LPTN-LA pathway, whereas the other study[34] reported an excitatory PV$^+$ SC-PBGN pathway to trigger escape followed by freezing. Data in the present study indicate that these two apparently redundant pathways co-exist in the mouse brain and differentially contribute to distinct defensive behavioral patterns triggered by looming visual stimuli.

In their early studies of the rodent SC, Redgrave and colleagues proposed that separate populations of SC neurons project to different downstream target areas, forming a "mosaic" cellular organization in the SC[26]. Our observation that two separate groups of SC PV$^+$ neurons independently project to the PBGN and LPTN seems to support this theory. Instead of using collaterals, these two groups of SC PV$^+$ neurons transmit similar looming visual signals to their downstream target nuclei. Although seemingly redundant, such design of neural circuits may facilitate the competition between PBGN and LPTN downstream of the SC.

A recent study[58] using AAV-mediated anterograde transsynaptic tagging showed that light stimulation of axon terminals in the PBGN to activate V1-innervated SC neurons did not produce escape behavior. This observation does not contradict with our results. Note that those authors have reported, in their earlier work[57], that activation of V1-innervated SC neurons triggered "temporary arrest behavior" without any escape. The absence of escape behavior after activation of V1-inervated SC neurons suggests that V1-innervated SC neurons may functionally differ from the SC PV$^+$ neurons.

An important observation in the present study is that the PBGN and LPTN neurons were co-activated during Type I and Type II behavioral patterns (Fig. 6). The lack of correlation between behavioral output and neural activity in these nuclei suggests a downstream target as a potential node of integration and behavioral choice. Previous studies have indicated that the LPTN projects to the lateral amygdala[39] and mediates visually triggered freezing[40]. The PBGN has been shown to project to the central amygdaloid nucleus (CeA)[34,59] and dorsal lateral periaqueductal gray (PAG)[60,61], both of which are involved in controlling defensive behaviors[62–65]. Therefore, it is possible that the PBGN and LPTN transmit the threat-relevant visual signals to the amygdala/PAG-associated network, which has been recently suggested as a central organizer for action selection between freezing and escape[62,63]. With the delineation of the hard-wired visual pathways between the SC and PBGN/LPTN, this possibility can now be tested.

## Methods

**Animals**. All experimental procedures were conducted following protocols approved by the Administrative Panel on Laboratory Animal Care at the National Institute of Biological Sciences and Institute of Biophysics, Chinese Academy of Sciences. *PV-ires-Cre*[66] and *vGlut2-ires-Cre* mice[67] were imported from the Jackson Laboratory (JAX Mice and Services). Mice were maintained on a circadian 12-h light/12-h dark cycle with food and water available ad libitum. Adult C57BL/6 mice (3–5-month-old) were housed in groups (3–5 animals per cage) before they were separated 1 week prior to virus injection. After virus injection, each mouse was housed in one cage for 3 weeks before subsequent experiments.

**Virus vector preparation**. The serotype of AAV is AAV-DJ[68]. The promoter to drive the expression of molecular tools is EF1a. Plasmids for AAV-DIO-EGFP and AAV-DIO-EGFP-2A-TeNT were gifts from Dr. Thomas Südhof (Stanford University). AAV-DIO-ChR2-mCherry was a gift from Dr. Karl Deisseroth (Stanford University). AAV-DIO-GCaMP6m and AAV-DIO-mCherry were constructed by replacing the coding region of ChR2-mCherry in AAV-DIO-ChR2-mCherry with those encoding mCherry or GCaMP6m (Addgene Plasmid 40754), respectively. AAV-DIO-EGFP-2A-NaChBac was constructed by replacing the coding region of TeNT

in AAV-DIO-EGFP-2A-TeNT with that encoding NaChBac, a gift from Dr. Benjamin White (Addgene plasmid # 40283). These viral particles were produced at Stanford Vector Core and at National Institute of Biological Sciences with help from Dr. Min-Min Luo. The final viral vector titers were in the range of $3-8 \times 10^{12}$ particles per ml.

**Stereotaxic injection of AAV and CTB.** Mice were anesthetized with intraperitoneal injection of tribromoethanol ($125-250$ mg kg$^{-1}$). Standard surgery was performed to expose the brain surface above the SC, PBGN or LPTN. Coordinates used for SC injection were: bregma $-3.80$ mm, lateral $\pm 0.50$ mm, and dura $-1.00$ mm. Coordinates used for PBGN injection were: bregma $-4.25$ mm, lateral $\pm 1.95$ mm, and dura $-2.60$ mm. Coordinates used for LPTN injection were: bregma $-2.06$ mm, lateral $\pm 1.35$ mm, and dura $-2.40$ mm. The AAV vectors and CTB-488/594 were stereotaxically injected with a glass pipette connected to Nano-liter Injector 201 (World Precision Instruments, Inc) at a slow flow rate of 0.15 μl min$^{-1}$ to avoid potential damage of local brain tissue. The pipette was withdrawn at least 20 min after viral injection.

For synaptic inactivation and increased excitability experiments (Figs. 2, 8, 9, and Supplementary Figs. 3, 8), all injections (AAV-DIO-EGFP, AAV-DIO-EGFP-2A-TeNT, AAV-DIO-EGFP-2A-NaChBac) were bilateral. Behavioral tests were conducted at 3 weeks after viral injection. For pathway tracing experiments (Fig. 3), the AAV (AAV-DIO-mGFP) and CTB injections were unilateral on the same side. Histological analyses were conducted 1 week (for CTB) and 3 weeks (for AAV) after injection. For fiber photometry (Figs. 4, 6) and optogenetic activation (Figs. 5, 7) experiments, the AAV injections (AAV-DIO-ChR2-mCherry, AAV-DIO-mCherry, AAV-DIO-GCaMP6m) were unilateral and followed by optic fiber implantation, as described below. All the injections were summarized in Supplementary Table 2.

**Optic fiber implantation.** Thirty minutes after AAV injections, a ceramic ferrule with an optic fiber (optogenetics: 200 μm in diameter, N.A. 0.22; fiber photometry: 230 μm in diameter, N.A. 0.37) was implanted with the fiber tip on top of the SC (bregma $-3.80$ mm, lateral $+0.50$ mm, and dura $-0.40$ mm), PBGN (bregma $-4.25$ mm, lateral $+1.95$ mm, and dura $-2.50$ mm), LPTN (bregma $-2.06$ mm, lateral $+1.35$ mm, dura $-2.20$ mm) and LDTN (bregma $-1.20$ mm, lateral $+1.00$ mm, dura $-2.00$ mm). The ferrule was then secured on the skull with dental cement. After implantation, the skin was sutured and antibiotics were applied to the surgical wound. The optogenetic and fiber photometry experiments were conducted at least 3 weeks after optic fiber implantation. All the optic fiber implantations were summarized in Supplementary Table 2.

Note that the optic fiber tip was usually 100~500 μm above the AAV injection center. For the SC, the optic fiber tip was ~500 μm above the injection center. For the PBGN, the optic fiber tip was 100 μm above the injection center. For the LPTN, the optic fiber tip was 200 μm above the injection center.

Given the small size of the PBGN (~300 μm in diameter), it was technically challenging to spatially target this nucleus with viral injection and with optic fiber implantation. The viral injection center and optic fiber track of each mouse was verified in the present study. Only mice with injection center localized within PBGN and mice with optic fiber tracks above the PBGN were used for the analyses. The mice without correct targeting of the PBGN were rejected without further analyses.

**Behavioral tests.** All behavioral tests were conducted during the same circadian period (1300–1900 hours). After virus injection or fiber implantation, the mice were housed individually for 3 weeks before the behavioral tests. They were handled daily by the experimenters for at least 3 days before the behavioral tests. On the day of the behavioral test, the mice were transferred to the testing room and were habituated to the room conditions for 3 h before the experiments started. The apparatus was cleaned with 20% ethanol to eliminate odor from other mice. All behaviors were scored by the experimenters, who were blind to the treatment of the animals.

Defensive behaviors triggered by looming visual stimuli were measured in an arena (35 cm × 35 cm square open field) with regular mouse bedding. No shelter was used. A regular computer monitor was positioned above the arena for presentation of overhead looming visual stimuli. After entering, the mice were allowed to explore the arena for 10 min. This was followed by the presentation of three cycles of overhead looming visual stimuli consisting of an expanding dark disk (Fig. 1b). The luminance of dark disk and background was 0.1 and 3.6 cd m$^{-2}$, respectively. Mouse behavior was recorded (25 fps) by two orthogonally positioned cameras with LEDs providing infrared illumination (Supplementary Fig. 1a). The location of the mouse in the arena (X, Y) was measured by a custom-written Matlab program using the formulae presented in Supplementary Fig. 1b. The instantaneous locomotion speed was calculated with a 200 ms time-bin.

To quantitatively describe the distinct behavioral patterns, we measured average locomotion speed before (3 s), peak speed during (3 s), and average speed after (15 s) looming visual stimuli and calculated the locomotion speed index of each mouse during stimuli (LSI$_{during stimuli}$) and after stimuli (LSI$_{after stimuli}$). LSI$_{during stimuli}$ was calculated as the log (base of 10) of the ratio between peak speed during stimuli and average speed before stimuli. LSI$_{after stimuli}$ was calculated as the log of the ratio between average speed after stimuli and that before stimuli. Defensive behavior with

a positive LSI$_{during stimuli}$ value and negative LSI$_{after stimuli}$ value was defined as escape-freezing pattern (Type I), whereas that with negative values for both LSI$_{during stimuli}$ and LSI$_{after stimuli}$ was defined as freezing-only pattern (Type II).

For optogenetically triggered defensive behaviors, a 473-nm diode pumped solid state laser system was used to generate the 473-nm blue laser for light stimulation. An FC/PC adaptor was used to connect the output of the laser to the implanted ferrule for intracranial light delivery. Before each experiment, the output of the laser was measured and adjusted to 20 mW. The mice were handled daily with all optics connected for at least 3 consecutive days before the behavioral test to reduce stress and anxiety. The pulse onset, duration, and frequency of light stimulation were controlled by a programmable pulse generator attached to the laser system. Locomotor behaviors were recorded with a camera above the home-cage and were analyzed with a video tracking system (Xeye Aba).

**Fiber photometry recording.** Fiber photometry[69] was used to record calcium transients from the cell bodies of PV$^+$ neurons in the SC and from their axon terminals in the PBGN and LPTN in anesthetized mice (Fig. 4). In addition, calcium transients from the cell bodies of glutamatergic neurons in the PBGN and LPTN were recorded in freely moving mice (Fig. 6).

For recording from anesthetized mice, 3 weeks before fiber photometry recording, AAV-DIO-GCaMP6m[70] was stereotaxically injected into the SC of *PV-ires-Cre* mice followed by optic fiber implantation above the SC, PBGN, or LPTN, as described above. On the day of recording, mice were anesthetized with urethane (20%, 1 ml per 100 g) and placed in a standard stereotaxic apparatus with the same orientation as when mice were at rest[34]. The body temperature was maintained at 37 °C using a heating pad. The contralateral eye was kept open, and the ipsilateral eye was covered to prevent viewing. A 45-cm wide and 35-cm high screen was placed 18 cm from the contralateral eye and 25° to the mid-sagittal plane of the mouse, resulting in a visually stimulated area (100° horizontal × 90° vertical) in the lateral visual field[34]. The orientation of the screen was adjusted ~45° to make the screen perpendicular to the eye axis of the contralateral eye (Fig. 4d).

After identification of the receptive field location on the screen of SC PV$^+$ neurons, a computer-generated soccer ball (diameter = 40 or 80 cm) moving toward the contralateral eye at different speeds in six directions (X+, X−, Y+, Y−, Z+, and Z−) was displayed on the receptive field. Black and white squares of equal area inside the soccer ball ensured that overall luminance was unchanged during its looming motion. The black and white square luminances were 0.1 and 6.6 cd m$^{-2}$, respectively. The soccer ball first appeared stationary on the screen for 2 s to collect baseline calcium signals as controls, and was then presented with an interval of at least 15 s between trials to allow the neurons to recover from any motion adaptation.

For recording from freely moving mice, 3 weeks before fiber photometry recording, AAV-DIO-GCaMP6m was stereotaxically injected into the PBGN and LPTN of *vGlut2-ires-Cre* mice, followed by optic fiber implantation above the PBGN or LPTN, as described above (Fig. 6c, d). On the day of recording, mice with optic fibers connected to the fiber photometry system were allowed to freely explore the arena (Fig. 6e). The looming visual stimuli were delivered to the contralateral eye while the calcium signals and defensive behaviors were simultaneously recorded. A flashing LED triggered by a 1-s square-wave pulse was simultaneously recorded to synchronize the video and calcium signals. The optic fiber tip sites above the PBGN and LPTN were carefully examined in each mouse after the experiments.

Fiber photometry system (ThinkerTech, Nanjing) was used for recording calcium signals from genetically identified neurons. To induce fluorescence signals, a laser beam from a laser tube (488 nm) was reflected by a dichroic mirror, focused by a ×10 lens (N.A. 0.3) and then coupled to an optical commutator (Fig. 4d). A 2-m optical fiber (230 μm in diameter, N.A. 0.37) guided the light between the commutator and implanted optical fiber. To minimize photo bleaching, the power intensity at the fiber tip was adjusted to 0.02 mW. The GCaMP6m fluorescence was band-pass filtered (MF525-39, Thorlabs) and collected by a photomultiplier tube (R3896, Hamamatsu). An amplifier (C7319, Hamamatsu) was used to convert the photomultiplier tube current output to voltage signals, which were further filtered through a low-pass filter (40 Hz cut-off; Brownlee 440). The analog voltage signals were digitalized at 100 Hz and recorded by a Power 1401 digitizer and Spike2 software (CED, Cambridge, UK).

**Slice physiology.** Brain slices containing the SC, PBGN, or LPTN were prepared from adult mice anesthetized with isoflurane before decapitation. Brains were rapidly removed and placed in ice-cold oxygenated (95% O$_2$ and 5% CO$_2$) cutting solution (228 mM sucrose, 11 mM glucose, 26 mM NaHCO$_3$, 1 mM NaH$_2$PO$_4$, 2.5 mM KCl, 7 mM MgSO$_4$, and 0.5 mM CaCl$_2$). Coronal brain slices (400 μm) were cut using a vibratome (VT 1200S, Leica Microsystems, Wetzlar, Germany). The slices were incubated at 28 °C in oxygenated artificial cerebrospinal fluid (ACSF: 119 mM NaCl, 2.5 mM KCl, 1 mM NaH$_2$PO$_4$, 1.3 mM MgSO$_4$, 26 mM NaHCO$_3$, 10 mM glucose, and 2.5 mM CaCl$_2$) for 30 min, and were then kept at room temperature under the same conditions for 1 h before transfer to the recording chamber at room temperature. The ACSF was perfused at 1 ml per min. The SC slices were visualized with a ×40 Olympus water immersion lens, differential interference contrast (DIC) optics (Olympus Inc., Japan), and a CCD camera (Q-Imaging Rolera-XR, BC, Canada).

Patch pipettes were pulled from borosilicate glass capillary tubes (Cat #64-0793, Warner Instruments, Hamden, CT, USA) using a PC-10 pipette puller (Narishige Inc., Tokyo, Japan). For recording of action potentials (current clamp), pipettes were filled with solution (in mM: 135 K-methanesulfonate, 10 HEPES, 1 EGTA, 1 Na-GTP, 4 Mg-ATP, and 2% neurobiotin, pH 7.4). Injected depolarizing currents were 0.1, 0.2, 0.3, 0.4 and 0.5 nA. Neuronal excitability was measured in the presence of PTX, D-APV and CNQX to block synaptic transmissions. For recording of postsynaptic currents (voltage clamp), pipettes were filled with solution (in mM, 135 CsCl, 10 HEPES, 1 EGTA, 1 Na-GTP, 4 Mg-ATP, pH 7.4). The resistance of pipettes varied between 3.0 and 3.5 MΩ. The current and voltage signals were recorded with MultiClamp 700B and Clampex 10 data acquisition software (Molecular Devices). After establishment of the whole-cell configuration and equilibration of the intracellular pipette solution with the cytoplasm, series resistance was compensated to 10–15 MΩ. Recordings with series resistances of >15 MΩ were rejected. An optic fiber (200 μm in diameter) was positioned above the brain slices, with laser intensity adjusted to variable powers (5, 10, 15, and 20 mW). Light-evoked action potentials from ChR2-mCherry$^+$ neurons were triggered by light-pulse train (473 nm, 1 ms, 10 Hz, 2~20 mW) synchronized with Clampex 10 data acquisition software (Molecular Devices). Light-evoked synaptic currents from ChR2-mCherry-negative SC neurons were triggered by single light pulses (1 ms) in the presence of 4-AP (20 μM) and TTX (1 μM). D-AP5 (50 μM)/CNQX (20 μM) or picrotoxin (PTX, 50 μM) were perfused with ACSF to examine the neurotransmitter type used by ChR2-mCherry-expressing neurons. Reagents for slice physiology are displayed in Supplementary Table 3.

**Histological procedures**. Mice were anesthetized with isoflurane and sequentially perfused with saline and phosphate buffered saline (PBS) containing 4% paraformaldehyde. Brains were removed and incubated in PBS containing 30% sucrose until they sank to the bottom. The post-fixation procedure of the brain was avoided to optimize immunohistochemistry of GABA and glutamate. Cryostat sections (40 μm) containing the SC, PBGN, and LPTN were collected, incubated overnight with blocking solution (PBS containing 10% goat serum and 0.7% Triton X-100), and then treated with primary antibodies diluted with blocking solution for 3–4 h at room temperature. Primary antibodies used for immunohistochemistry are displayed in Supplementary Table 3. Primary antibodies were washed three times with washing buffer (PBS containing 0.7% Triton X-100) before incubation with secondary antibodies (tagged with Cy2, Cy3, or Cy5; dilution 1:500; Life Technologies Inc., USA) for 1 h at room temperature. Sections were again washed three times with washing buffer, stained with DAPI in PBS, transferred onto Super Frost slides, and mounted under glass coverslips with mounting media.

Sections were imaged with an Olympus VS120 epifluorescence microscope (×10 objectives) or an Olympus FV1200 laser scanning confocal microscope (×20 and ×60 objectives). Samples were excited by 488, 543, or 633 nm lasers in sequential acquisition mode to avoid signal leaking. Saturation was avoided by monitoring pixel intensity with Hi-Lo mode. Confocal images were analyzed with ImageJ Software.

**Cell-counting strategy**. The cell-counting strategy was summarized in Supplementary Table 1. For counting cells in the SC, we collected all the 40-μm coronal sections from Bregma −3.28 to Bregma −4.48 for each mouse. Then six sections that were evenly spaced by 200 μm were sampled for immunohistochemistry to label cells positive for different markers. We acquired confocal images (Olympus FV1200 microscope, ×20 objective) within SuG and Op layers of the SC followed by cell counting with ImageJ. We calculated the percentage of glutamate$^+$ and GABA$^+$ neurons in SC PV$^+$ neuronal populations (Fig. 2a). We also calculated the percentage of PV$^+$ neurons in neuronal population labeled by different molecular tools and vice versa (CTB-488 and CTB-594: Fig. 3h; EGFP-2A-TeNT: Supplementary Fig. 3b; mGFP: Supplementary Fig. 4a; GCaMP6m: Supplementary Fig. 5a; ChR2-mCherry: Supplementary Fig. 6b).

For counting cells in the PBGN, we collected all the five coronal sections (40 μm) from Bregma −4.16 to Bregma −4.36 that contain PBGN. These sections were then used for immunohistochemistry to label PBGN cells positive for different markers. For counting cells in the LPTN, we collected all the coronal sections (40 μm) from Bregma −1.82 to Bregma −2.80 for each mouse. Then five sections that were evenly spaced by 200 μm were sampled for immunohistochemistry to label LPTN cells positive for different markers. After image acquisition, we outlined the PBGN and LPTN, followed by cell counting with Image J. We calculated the percentage of glutamate$^+$ and GABA$^+$ neurons in NeuN$^+$ populations (Supplementary Fig. 7a, b). We also calculated the percentage of glutamate$^+$ neurons in neuronal population labeled by different molecular tools and vice versa (EGFP: Supplementary Fig. 7c, d; GCaMP6m: Supplementary Fig. 7e, f; ChR2-mCherry: Supplementary Fig. 7g, h; EGFP-2A-TeNT: Supplementary Fig. 7i, j).

**Data quantification and statistical analyses**. All experiments were performed with anonymized sample in which the experimenter was unaware of the experimental condition of mice. Student's *t* test was used to analyze behavioral data (Figs. 2, 5, 7, 8, and Supplementary Fig. 6), fiber photometry data (Figs. 4, 6, and Supplementary Fig. 5), slice physiology data (Fig. 7, Supplementary Figs. 3 and 6) and morphological data (Fig. 3). The one-way ANOVA was used to analyze input−output curves of PSCs (Figs. 2, 9, Supplementary Figs. 3 and 8). The

sample size "*n*" represents number of mice (behavioral, morphological, and fiber photometry analyses) or number of cells (slice physiological analyses). For the behavioral analyses, the sample size is 6~25 mice, depending on the specific aim of the experiment. For the fiber photometry analyses, the sample size is 6~9 mice. For morphological analyses, the sample size is 3~5 mice. For the slice physiological analyses, the sample size is 6~12 cells. No randomization was used. The variance between the groups that are being statistically compared is similar. For every figure, statistical tests are justified as appropriate. The data meet the normal distribution. There is an estimate of variation within each group of data.

**Data availability**. The data that support the findings of this study are available from the corresponding author upon reasonable request.

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

## Acknowledgements

We appreciate the generosity of Drs. Karl Deisseroth and Thomas Südhof for providing plasmids for this study. We thank Yan Teng, Lili Niu, Xudong Zhao and Qifei Jiang for technical support. This work was supported by the National Natural Science Foundation of China (31671095, 31422026, 81471311, and 91632301 to P.C.; 31771150 to Y.W.) and Startup Funding Support at National Institute of Biological Sciences, Beijing.

## Author contributions

P.C. conceived the study. C.S. A.L., F.Y., Y.Z., and W.L. did injections and fiber implantation. Z.C. and Z.L. did slice physiology. Z.C., C.S., A.L., Y.L., J.Z., B.Q., F.Y., Y.Z., and W.L. did behavioral tests. C.S., Y.L., and J.Z. did fiber photometry recording. Z.C., C.S., A.L., J.Z., B.Q., Y.L., Y.Z., and W.L. did histological analyses. Y.W. and X.G. provide instruments. D.L., C.S., Z.C., and P.C. analyzed data. P.C. wrote the manuscript.

## Additional information

**Competing interests:** The authors declare no competing interests.

