## [Peer Review File(PDF 426 kb) · Nature Communications]

Reviewers' comments:

Reviewer #1 (Remarks to the Author):

In their manuscript "Divergent midbrain circuits to orchestrate innate defensive behaviors in mice", Peng Cao and colleagues demonstrate that dimorphic defensive behavioral patterns can emerge downstream from the activation of parvalbumin-positive (PV+) neurons of the superior colliculus (SC). Through a series of elegant experiments, the authors demonstrate that independent populations of PV+ neurons of the SC differentially innervate the parabrachial nucleus (PBGN) and the lateral posterior thalamic nucleus (LPTN), and that this differential innervation participates in the generation of dimorphic defensive behaviors. While I find that the data presented here is beautiful, and that the experimental design is logical and rigorous I am failing to see the degree to which the study is novel. For example, this study is a follow up to a seminal manuscript published in 2015 by the same authors. In that manuscript, the authors had already identified PV+ neurons of the SC as a driver of defensive behaviors, and also pointed out to their projections to the PBGN as causally involved in the generation of escape-then-freeze behavior. Notably, a few months before their 2015 study was published, another manuscript published in Nature Communications demonstrated that activating projections from the SC to the LPTN triggered freezing responses (Wei et al., 2015), an argument made by the present manuscript. Therefore, one could argue that the idea (and inferably the evidence) that projections of the SC to two independent targets participate in the generation of dimorphic defensive behaviors is not novel. Early in their abstract, the authors argue "The neural circuit mechanisms underlying behavioral choice between escape and freezing remain unclear" suggesting this as the focus of the present study. However, the authors never demonstrated what those neural circuit mechanisms that underlie behavioral choice are. Instead, the major finding of the present study is that, surprisingly, the SC/PV+ input to the PBGN and LPTN is active regardless of behavioral choice. This suggests that behavioral choice is made elsewhere (e.g. the amygdala). Therefore, given that many of the observations made here largely reproduce the two 2015 publications on this topic, it appears to me that the authors failed to show the neural circuit mechanisms by which dimorphic behaviors arise. In fact, I am surprised that the authors did not choose PBGN and LPTN as a starting point towards addressing this important question, and spent half of their manuscript essentially confirming previous observations.

Minor points:

- 1) The controls that the authors did for their TeNT experiments in which they tested the efficacy of this approach to abolish synaptic transmission are mostly adequate. However, the authors should consider that demonstrating that ChR2 is still able to elicit action potential on infected neurons that also express the TeNT construct in slices used for synaptic transmission assessment is probably a better control. This will confirm that the reduction in EPSC responses are in fact due to an impairment in synaptic transmission.
- 2) The authors should consider rephrasing the beginning of their abstract. It gives the impression that the work seeks to define "The neural circuit mechanisms underlying behavioral choice between escape and freezing".
- 3) In their manuscript's discussion, the authors need to speculate on why there are two separate populations of PV+ neurons in SC that project to PBGN and LPTN. If they are essentially relaying the same kind of information to downstream targets and do not seemingly partake in behavioral choice, why not collaterals?

Reviewer #2 (Remarks to the Author):

This manuscript examines pathways originating from the superior colliculus (SC) in different forms of defensive behaviors. Using an impressive array of approaches (behavior, optogenetics, tetanus neurotoxin, fiber photometry, slice physiology) to record, activate and inactivate cell-type and projection-specific neurons, the authors make a strong case that the pathway from (primarily glutamatergic) parvalbumin-positive (PV+) SC neurons to the PBGN mediates an escape-freezing response, while the pathway from PV+ SC neurons to LPTN mediates a freezing-only response, to looming visual stimuli. It is particularly impressive that experiments are performed at multiple stages of the pathway: PV+ SC neurons are shown to be critical for both behavioral responses, and their downstream targets (in PBGN and LPTN) are shown to be critical for one of the responses. While how these targets “compete” to determine the behavioral response is unknown, this important question can now be addressed given the findings of this paper.

The experiments appear to be well-controlled and interpretable and the analyses are straightforward, so the conclusions appear to be justified. In general this is a great study with significant findings that should have an impact on the field. I have only one important concern, about the behavioral paradigm itself.

Are the behavioral data (e.g., in Fig. 1, but the question applies to many other figures as well) from a single trial of the behavior (in which each trial consisted of 3 repeats of the looming visual stimulus)? Or did each mouse perform multiple trials, and the data shown are the average across trials?

If the latter (multiple trials/mouse), was it the case that a given mouse always showed the same pattern of behavior (Type 1 or 2)? Why would some mice consistently exhibit “Type 1” and some consistently exhibit “Type 2” behavior? It would be interesting to speculate about this, perhaps in the context of the subsequent findings about the roles of PBGN and LPTN in these two behaviors.

If the former (1 trial/mouse), then of course it isn't known whether there are “Type 1” mice and “Type 2” mice, but it would again be interesting to speculate whether it would be the case. More importantly, though, some explanation of why only 1 trial/mouse was performed would be helpful. It would seem that repeating the experiment several times would be beneficial to the analyses. Is adaptation to the stimulus a concern?

Minor questions/concerns:

Given that several downstream targets of PV+ SC neurons were identified (Supplementary Fig. 4), some further justification for focusing here on PBGN and LPTN would be useful. Do the authors think the other pathways are also playing roles in defensive behavior?

What are the dashed lines in Fig 1h,i?

Does Fig. 4n show the data for only 1 diameter-velocity combination?

Authors' response to the referees' comments for Shang et al., "**Divergent midbrain circuits to orchestrate innate defensive behaviors in mice**", and changes made in the revised manuscript

We really appreciate the referees' careful evaluation of our manuscript. We thank them for their positive and constructive comments, which are very helpful for us to revise the manuscript. We are now submitting the revised manuscript to fully address all the referees' concerns. During the revision, we have performed a series of additional experiments. Specifically, we have:

1. Further dissected circuit mechanism underlying freeze-or-flight action selection, by examining the effects of increasing intrinsic excitability of PBGN or LPTN neurons on behavioral outcome to looming visual stimuli (Fig. 9). At the end of Results, we plotted a summarized diagram to illustrate the circuit mechanism underlying freeze-or-flight action selection revealed in the present study (Fig. 10).
2. Recorded the light-evoked action potential firing from neurons co-expressing ChR2-mCherry and TeNT, and found that the light-evoked action potential firing was not impaired by TeNT expression (Supplementary Fig. 3, **m - o**).
3. Examined the behavioral responses of WT mice to multiple trials of looming visual stimuli (Supplementary Fig. 1**c, d**).
4. Analyzed the fiber photometry data recorded from axon terminals of SC PV⁺ neurons in response to the virtual looming soccer ball with the other three "diameter-velocity" combinations (Supplementary Fig. 5, **f - h**).

We hope that with these additions and the changes introduced into the manuscript, it can now be accepted for publication. In the following, we cite the referees' comments in full in *italic* typeface, and then provide our answers in **bold** typeface.

Referee #1 :

In their manuscript "Divergent midbrain circuits to orchestrate innate defensive behaviors in mice", Peng Cao and colleagues demonstrate that dimorphic defensive behavioral patterns can emerge downstream from the activation of parvalbumin-positive (PV+) neurons of the superior colliculus (SC). Through a series of elegant experiments, the authors demonstrate that independent populations of PV+ neurons of the SC differentially innervate the paraventricular nucleus (PBGN) and the lateral posterior thalamic nucleus (LPTN), and that this differential innervation participates in the generation of dimorphic defensive behaviors.

We very much appreciate the referee's comments, which are mostly helpful and may partly reflect our weak clarification in the original manuscript. We hope we have

addressed the referee's concerns in the revised manuscript as described below, and that the referee will be satisfied with our explanations and new experimental data.

While I find that the data presented here is beautiful, and that the experimental design is logical and rigorous, I am failing to see the degree to which the study is novel. For example, this study is a follow up to a seminal manuscript published in 2015 by the same authors. In that manuscript, the authors had already identified PV+ neurons of the SC as a driver of defensive behaviors, and also pointed out to their projections to the PBGN as causally involved in the generation of escape-then-freeze behavior. Notably, a few months before their 2015 study was published, another manuscript published in Nature Communications demonstrated that activating projections from the SC to the LPTN triggered freezing responses (Wei et al., 2015), an argument made by the present manuscript. Therefore, one could argue that the idea (and inferably the evidence) that projections of the SC to two independent targets participate in the generation of dimorphic defensive behaviors is not novel.

We agree with the referee that the present study follows up to our earlier work (Shang et al., 2015). However, we did not seek to reproduce the old findings. Instead, we asked how distinct defensive behaviors are generated and selected by the brain. Specifically, the present study addresses three important questions in the field of innate defensive behavior. First, we have examined the role of SC PV⁺ neurons in visually triggered defensive behaviors (Fig. 1 and Fig. 2), a key question that remained unanswered in the earlier work (Shang et al., 2015). Second, we have addressed an apparent discrepancy between two recent studies in the field of innate defensive behavior. A paper published in *Nature Communications* (Wei et al., 2015) reported that activating projections from the SC to the LPTN triggered freezing responses. In our work (Shang et al., 2015), we identified PV⁺ SC-PBGN pathway that causally drive the escape-then-freeze behavior. Naturally, these two groups have the responsibility to clarify the apparent discrepancy. The data in the present study indicate that these two apparently redundant pathways (SC-PBGN and SC-LPTN) co-exist in mouse brain and differentially contribute to distinct defensive behavioral patterns (Fig. 3, Fig. 4 and Fig. 5). Third, we have explored how SC-PBGN and SC-LPTN work together to determine the action selection (Fig. 6, Fig. 7 and Fig. 8). Based on the results, we proposed a “winner-take-all” model as a circuit mechanism for the action selection.

With the answers to the above three questions, we concluded that SC orchestrates dimorphic defensive behaviors with two divergent excitatory visual pathways that work in a “winner-take-all” mode. This conclusion, which has not been reached previously in this field, may have extended our concept on how distinct defensive behavioral patterns are generated and selected by the brain.

We hope the referee will concur with this overall assessment. We apologize for not having expressed this clearly in the original manuscript.

Early in their abstract, the authors argue “The neural circuit mechanisms underlying behavioral choice between escape and freezing remain unclear” suggesting this as the focus of the present study. However, the authors never demonstrated what those neural circuit mechanisms that underlie behavioral choice are. Instead, the major finding of the present study is that, surprisingly, the SC/PV+ input to the PBGN and LPTN is active regardless of behavioral choice. This suggests that behavioral choice is made elsewhere (e.g. the amygdala). Therefore, given that many of the observations made here largely reproduce the two 2015 publications on this topic, it appears to me that the authors failed to show the neural circuit mechanisms by which dimorphic behaviors arise. In fact, I am surprised that the authors did not choose PBGN and LPTN as a starting point towards addressing this important question, and spent half of their manuscript essentially confirming previous observations.

We very much appreciate the referee’s comments, which are very helpful and reflect our weak clarification in the original manuscript. We agree with the referee that in the abstract “the neural circuit mechanisms underlying behavioral choice between escape and freezing remain unclear” may be misleading to some extent. In the abstract of revised manuscript, we replace the old sentence with “the key neural circuits that participate in the generation of dimorphic defensive behaviors remain unclear” (Page 2, Line 22-23).

To seek for the circuit mechanisms by which dimorphic defensive behaviors are generated, we actually have provided data (Fig. 6 and Fig. 8) that support a “winner-take-all” model in the original manuscript. We sincerely apologize for not having clarified this model clearly. We showed that both PBGN neurons and LPTN neurons are active in dimorphic defensive behaviors (Fig. 6). This interesting observation raised a hypothesis that SC-PBGN and SC-LPTN pathways may be actively competing for the behavioral outcome to looming visual stimuli, namely “winner-take-all” hypothesis. In Fig. 8, we tested this hypothesis by examining the effects of selective silencing of PBGN or LPTN neurons on visually triggered defensive behaviors. Strikingly, bilateral inactivation of one nucleus resulted in the behavioral outcome dominated by the other nucleus, a finding supporting the “winner-take-all” hypothesis.

During the revision of the manuscript, we performed additional experiments to further test the “winner-take-all” hypothesis. We selectively increased the intrinsic excitability of PBGN or LPTN neurons, and examined the effects on visually triggered defensive behaviors (Fig. 9 in the revised manuscript). NaChBac, a bacterial voltage-dependent sodium channel, has an activation threshold ~ 15 mV more negative than that of native sodium channel in neuron (Ren et al., 2001). This property has made it a molecular tool to manipulate neuronal excitability in both drosophila (Luan et al., 2006) and mice (Kelsch et al., 2009; Lin et al., 2010). We bilaterally injected AAV-DIO-EGFP-2A-NaChBac into the PBGN or LPTN of *vGlut2-ires-Cre* mice, resulting in selective expression of NaChBac, as reported by EGFP, in PBGN and LPTN neurons (Fig. 9a, b). The efficiency to increase neuronal excitability by NaChBac was confirmed in slice physiology (Fig.9 c-g). Interestingly, selective increase of neuronal excitability in the PBGN by NaChBac expression increased the proportion of mice with type I behavioral pattern (From 66.7% to 100%; Fig.9 h-j). On

the other hand, NaChBac expression in bilateral LPTN increased the proportion of mice with type II behavioral pattern (From 26.7% to 60%; Fig.9 k-m). The new data in Fig. 9 indicated that “gain-of-function” of PBGN or LPTN neurons can shift the behavioral outcome to the pattern dominated by the same nucleus. Therefore, data in Fig. 8 (loss-of-function) and in Fig. 9 (gain-of-function) together suggested that the SC-PBGN and SC-LPTN pathways form two mutually competing circuit modules to determine the outcome of dimorphic defensive behavioral patterns to looming visual stimuli. These new data have been added to Fig. 9 and are described in the revised manuscript (Page 19-20, Line 342-366).

At the end of the Results, we plotted a summarized diagram to illustrate the circuit mechanism underlying the action selection between dimorphic defensive behaviors that was revealed in the present study (Fig. 10). Please see the legend of Fig.10 in the revised manuscript (Page 58, Line 1063-1070).

Minor points:

1) The controls that the authors did for their TeNT experiments in which they tested the efficacy of this approach to abolish synaptic transmission are mostly adequate. However, the authors should consider that demonstrating that ChR2 is still able to elicit action potential on infected neurons that also express the TeNT construct in slices used for synaptic transmission assessment is probably a better control. This will confirm that the reduction in EPSC responses is in fact due to impairment in synaptic transmission.

We thank the referee for the great suggestion. We have performed additional experiments in acute SC slices showing that ChR2 is still able to reliably elicit action potential from neurons co-expressing TeNT and ChR2-mCherry. These new analyses have been added to Supplementary Fig. 3 and are described in the main text of the revised manuscript (Page 8, Line 129-132).

2) *The authors should consider rephrasing the beginning of their abstract. It gives the impression that the work seeks to define “The neural circuit mechanisms underlying behavioral choice between escape and freezing”.*

We thank the referee for this constructive comment. In the revised manuscript, we have rephrased the second sentence of the abstract, which now is “ the key neural circuits that participate in the generation of dimorphic defensive behaviors remain unclear” (Page 2, Line 22-23).

3) *In their manuscript’s discussion, the authors need to speculate on why there are two separate populations of PV+ neurons in SC that project to PBGN and LPTN. If they are essentially relaying the same kind of information to downstream targets and do not seemingly partake in behavioral choice, why not collaterals?*

We thank the referee for the constructive comment. We have added a paragraph to discuss about the two separate populations of PV⁺ neurons in the SC that project to PBGN and LPTN (Page 23-24, Line 420-427).

Referee #2 :

This manuscript examines pathways originating from the superior colliculus (SC) in different forms of defensive behaviors. Using an impressive array of approaches (behavior, optogenetics, tetanus neurotoxin, fiber photometry, slice physiology) to record, activate and inactivate cell-type and projection-specific neurons, the authors make a strong case that the pathway from (primarily glutamatergic) parvalbumin-positive (PV+) SC neurons to the PBGN mediates an escape-freezing response, while the pathway from PV+ SC neurons to LPTN mediates a freezing-only response, to looming visual stimuli. It is particularly impressive that experiments are performed at multiple stages of the pathway: PV+ SC neurons are shown to be critical for both behavioral responses, and their downstream targets (in PBGN and LPTN) are shown to be critical for one of the responses. While how these targets “compete” to determine the behavioral response is unknown, this important question can now be addressed given the findings of this paper.

We thank the referee very much for the very positive comments.

The experiments appear to be well-controlled and interpretable and the analyses are straightforward, so the conclusions appear to be justified. In general this is a great study with significant findings that should have an impact on the field. I have only one important concern, about the behavioral paradigm itself.

We thank the referee very much for the very positive comments.

Are the behavioral data (e.g., in Fig. 1, but the question applies to many other figures as well) from a single trial of the behavior (in which each trial consisted of 3 repeats of the looming visual stimulus)? Or did each mouse perform multiple trials, and the data shown are the average across trials?

We thank the referee and really appreciate the comment. Yes, the behavioral data in all Figures are from a single trial of the behavior, with the trial consisting of 3 repeats of looming visual stimulus. The experiments were designed in this way, because we would like to avoid behavioral adaptation caused by multiple trials. To make this point clear, we have added one sentence in the main text of the revised manuscript (Page 6, Line 91 - 92).

If the latter (multiple trials/mouse), was it the case that a given mouse always showed the same pattern of behavior (Type 1 or 2)? Why would some mice consistently exhibit “Type 1” and some consistently exhibit “Type 2” behavior? It would be interesting to speculate about this, perhaps in the context of the subsequent findings about the roles of PBGN and LPTN in these two behaviors.

If the former (1 trial/mouse), then of course it isn't known whether there are “Type 1” mice and “Type 2” mice, but it would again be interesting to speculate whether it would be the case. More importantly, though, some explanation of why only 1 trial/mouse was performed would be helpful. It would seem that repeating the experiment several times would be beneficial to the analyses. Is adaptation to the stimulus a concern?

We thank the referee and very much appreciate the constructive comments. In the original manuscript, we used the former paradigm (1 trial / mouse). We agree with the referee that it would be interesting to find out whether there are “Type 1” and “Type 2” mice. During the revision of the manuscript, we performed the latter paradigm (3 trials / mouse) suggested by the referee. Interestingly, mice with type 1 pattern in the first trial showed type 1 pattern in the second and third trials. However, the LSI _{during stimuli} and LSI _{after stimuli} rapidly declined. Mice with type 2 pattern in the first trial showed type 2 pattern in the second and third trials, with LSI _{during stimuli} and LSI _{after stimuli} rapidly declined as well. Together, these new data indicate that a given mouse showed the same pattern of behavior, either Type 1 or 2, in multiple trials, and their defensive behaviors rapidly adapt to the stimulus. These new data have been added to Supplementary Fig. 1, and are described in the main text of the revised manuscript (Page 6, Line 87-91).

Then why do some mice consistently exhibit “Type 1” and some mice consistently exhibit “Type 2” behavior? In the context of the findings about the roles of PBGN and LPTN in these two behaviors, we have the following speculations. The mice exhibiting “Type 1” behavior in multiple trials may possess physiologically stronger SC-PBGN pathways in weight than SC-LPTN pathways. The mice exhibiting “Type 2” behavior in multiple trials may have stronger SC-LPTN pathways in weight than SC-PBGN pathways. We have added these speculations to the discussion of the revised manuscript (Page 22, Line 392-397).

Minor questions/concerns:

Given that several downstream targets of PV+ SC neurons were identified (Supplementary Fig. 4), some further justification for focusing here on PBGN and LPTN would be useful. Do the authors think the other pathways are also playing roles in defensive behavior?

We thank the referee and appreciate the very helpful suggestion. We have discussed about the other pathways formed by PV⁺ SC neurons that may be

indirectly involved in defensive behaviors. A discussion has been added to the main text of the revised manuscript (Page 24, Line 436 - 440)

What are the dashed lines in Fig 1h,i?

We thank the referee for the constructive comment. We apologize for not having explained this in figure legend of the original manuscript. The dashed lines in Fig. 1h, i indicate the averaged locomotion speed before looming visual stimuli. A note has been added to the legend of Fig.1 in the revised manuscript (Page 49, Line 911 - 912).

Does Fig. 4n show the data for only 1 diameter-velocity combination?

We thank the referee and really appreciate the constructive comment. In the revised manuscript, we have analysed the fiber photometry data for the other three diameter-velocity combinations and presented these analyses in Supplementary Fig. 5 f-h.

The authors have diligently addressed my earlier questions. I appreciate their candid responses. I don't see any major issues that preclude publication.

We thank the referee very much, and really appreciate the positive comment.

Cited references:

1. Shang, C. et al. BRAIN CIRCUITS. A parvalbumin-positive excitatory visual pathway to trigger fear responses in mice. *Science* 348, 1472-1477 (2015).
2. Wei, P. et al. Processing of visually evoked innate fear by a non-canonical thalamic pathway. *Nat Commun* 6, 6756 (2015).
3. Ren, D. et al. A prokaryotic voltage-gated sodium channel. *Science* 294, 2372-2375, (2001).

4. Luan, H. et al. Functional dissection of a neuronal network required for cuticle tanning and wing expansion in *Drosophila*. *The Journal of Neuroscience* 26, 573-584, (2006).
5. Kelsch, W., Lin, C. W., Mosley, C. P. & Lois, C. A critical period for activity-dependent synaptic development during olfactory bulb adult neurogenesis. *The Journal of Neuroscience* 29, 11852-11858, (2009).
6. Lin, C. W. et al. Genetically increased cell-intrinsic excitability enhances neuronal integration into adult brain circuits. *Neuron* 65, 32-39, (2010).

Reviewers' comments:

Reviewer #1 (Remarks to the Author):

In the revised version of their manuscript "Divergent midbrain circuits to orchestrate innate defensive behaviors in mice", Peng Cao and colleagues performed additional experiments that improved the quality of their manuscript. In particular, they added new evidence that highlights the ability of PBGN and LPTN to select for type I and type II defensive responses, respectively. This evidence shows that increasing the excitability of neurons in either one of these nuclei mimics the results previously obtained with ChR2 stimulation. Whereas their approach here is quite elegant, I am failing to see how this manipulation improves the paper. The reason for this is that this manipulation produces an artificial increase in neuronal excitability that resembles the effect of ChR2 stimulation. Therefore, in light of the data already presented in the original version of this manuscript, it is not surprising that biasing the activity of one nucleus over the other in this manner also biases behavior. So, the authors have yet to support their conclusion that their data "reveal a winner take all" circuit mechanism experimentally.

The main issue here is that the authors' assessment of the stimulus evoke activity in the PBGN and LPTN revealed that these structures are equally active during type I and type II responses. As a result, it appears that there is no correlation between neural activity in these nuclei and behavioral output. So in the end one can conclude that although the activity of PBGN and LPTN can certainly bias behavioral selection, and likely participate in a winner-take-all process, under normal circumstances it is likely that behavioral selection occurs downstream. For example, neuromodulatory systems may bias the impact of the output of either one of these two regions and support selection, an outcome that could be overcome by artificial activation with optogenetics or by increasing neuronal excitability artificially with a viral construct (as done by the authors). In conclusion, it is this reviewer's humble opinion that the authors' affirmation of their data revealing a winner take all circuit mechanism is rather premature.

That said, I still think that the work presented here is important and improves the field by: 1) showing that PBGN and LPTN can bias behavioral selection, and 2) pointing at a downstream target as a potential node of integration and behavioral selection. However, I am recommending that the authors soften their stance in their abstract and manuscript in general by clarifying that their data "suggests/indicates" that a winner take all model is likely used here, but that this is simply a parsimonious conclusion based on current data and not on definite proof. I understand that achieving a holistic picture of the mechanisms underlying behavioral selection of defensive responses is a challenging task. But, without conclusive evidence one needs to acknowledge when presented evidence is suggestive of a particular model and not proof of it.

Reviewer #2 (Remarks to the Author):

The authors have addressed my previous concerns, and I have no further concerns.

Authors' response to the referees' comments for Shang et al., "**Divergent midbrain circuits to orchestrate innate defensive behaviors in mice**", and changes made in the revised manuscript

We really appreciate the referees' careful evaluation of our manuscript. We thank them for their positive and constructive comments, which are very helpful for us to revise the manuscript. We are now submitting the revised manuscript to fully address the referees' concerns. All changes in the revised manuscript have been highlighted in the manuscript text file. We hope that with the changes introduced into the manuscript, it can now be accepted for publication. In the following, we cite the referees' comments in full in *italic* typeface, and then provide our answers in **bold** typeface.

Referee #1 :

In the revised version of their manuscript "Divergent midbrain circuits to orchestrate innate defensive behaviors in mice", Peng Cao and colleagues performed additional experiments that improved the quality of their manuscript. In particular, they added new evidence that highlights the ability of PBGN and LPTN to select for type I and type II defensive responses, respectively. This evidence shows that increasing the excitability of neurons in either one of these nuclei mimics the results previously obtained with ChR2 stimulation. Whereas their approach here is quite elegant, I am failing to see how this manipulation improves the paper. The reason for this is that this manipulation produces an artificial increase in neuronal excitability that resembles the effect of ChR2 stimulation. Therefore, in light of the data already presented in the original version of this manuscript, it is not surprising that biasing the activity of one nucleus over the other in this manner also biases behavior. So, the authors have yet to support their conclusion that their data "reveal a winner take all" circuit mechanism experimentally.

The main issue here is that the authors' assessment of the stimulus evoke activity in the PBGN and LPTN revealed that these structures are equally active during type I and type II responses. As a result, it appears that there is no correlation between neural activity in these nuclei and behavioral output. So in the end one can conclude that although the activity of PBGN and LPTN can certainly bias behavioral selection, and likely participate in a winner-take-all process, under normal circumstances it is likely that behavioral selection occurs downstream. For example, neuromodulatory systems may bias the impact of the output of either one of these two regions and support selection, an outcome that could be overcome by artificial activation with optogenetics or by increasing neuronal excitability artificially with a viral construct (as done by the authors). In conclusion, it is this reviewer's humble opinion that the authors' affirmation of their data revealing a winner take all circuit mechanism is rather premature.

That said, I still think that the work presented here is important and improves the field by: 1) showing that PBGN and LPTN can bias behavioral selection, and 2) pointing at a downstream target as a potential node of integration and behavioral selection. However, I am recommending that the authors soften their stance in their abstract and manuscript in

general by clarifying that their data “suggests/indicates” that a winner take all model is likely used here, but that this is simply a parsimonious conclusion based on current data and not on definite proof. I understand that achieving a holistic picture of the mechanisms underlying behavioral selection of defensive responses is a challenging task. But, without conclusive evidence one needs to acknowledge when presented evidence is suggestive of a particular model and not proof of it.

We very much appreciate the referee’s comments. We agree with the referee that achieving a holistic picture of the “winner-take-all” model needs much more efforts. At this point, this model is only suggested but not proved. The word “reveal” in the original manuscript is not appropriate. We completely accept the referee’s recommendation that “the authors soften their stance in their abstract and manuscript in general by clarifying that their data “suggests/indicates” that a winner take all model is likely used here”. Accordingly, we introduced four changes in the revised manuscript, which have been highlighted in the manuscript text file.

First, in the Abstract (Page 2, Line 31-33), we replace the old last sentence with a new one: “Together, these data suggest that the SC orchestrates dimorphic defensive behaviors with two mutually competitive tectofugal pathways that likely participate in a winner-take-all process”.

Second, in the last paragraph of the Introduction, we replace the old sentence with a new one “Our data indicate that the SC PV⁺ neurons orchestrate these dimorphic defensive behaviors with two divergent tectofugal visual pathways” (Page 4, Line 60-61).

Third, in the first paragraph of the Discussion, we replace the old sentence with a new one “Together, these data suggest a dual-circuit winner-take-all mechanism that might be used by the SC to orchestrate dimorphic defensive behaviors in mammals” (Page 21, Line 375-377).

Finally, in the last paragraph of the Discussion, we point out the lack of correlation between behavioral output and neural activity in PBGN/LPTN. We clarify that this observation suggests a downstream target as a potential node of integration and behavioral choice (Page 24, Line 425-436).

We hope the referee will concur with us and be satisfied with the changes that we made in the revised manuscript.

Referee #2 :

The authors have addressed my previous concerns, and I have no further concerns.

We really appreciate the referee's careful evaluation of our manuscript. We thank her/him for the positive and constructive comments, which are very helpful for us to revise the manuscript.